# Discover and Align Taxonomic Context Priors for Open-world Semi-Supervised Learning

**Yu Wang**[1,6] **Zhun Zhong**[2] **Pengchong Qiao**[1,6] **Xuxin Cheng**[1] **Xiawu Zheng**[3,4]

**Chang Liu**[5] **Nicu Sebe**[7] **Rongrong Ji**[3,4] **Jie Chen**[1,3,6*]

[1]School of Electronic and Computer Engineering, Peking University, Shenzhen, China
[2]School of Computer Sceince, University of Nottingham, United Kingdom
[3]Peng Cheng Laboratory, Shenzhen, China    [4] Key Laboratory of Multimedia Trusted
Perception and Efficient Computing, Ministry of Education of China, Xiamen University
[5] Department of Automation, Tsinghua University, Beijing, China
[6] AI for Science (AI4S)-Preferred Program, Peking University Shenzhen Graduate School, China
[7]Department of Information Engineering and Computer Science, University of Trento, Italy

{rain_wang, pcqiao, chengxx}@stu.pku.edu.cn    {zhengxw01, chenj}@pcl.ac.cn    zhunzhong007@gmail.com
rrji@xmu.edu.cn    liuchang2022@tsinghua.edu.cn    niculae.sebe@unitn.it

## Abstract

Open-world Semi-Supervised Learning (OSSL) is a realistic and challenging task, aiming to classify unlabeled samples from both seen and novel classes using partially labeled samples from the seen classes. Previous works typically explore the relationship of samples as priors on the pre-defined single-granularity labels to help novel class recognition. In fact, classes follow a taxonomy and samples can be classified at multiple levels of granularity, which contains more underlying relationships for supervision. We thus argue that learning with single-granularity labels results in sub-optimal representation learning and inaccurate pseudo labels, especially with unknown classes. In this paper, we take the initiative to explore and propose a uniformed framework, called **T**axonomic context pr**I**ors **D**iscovering and **A**ligning (TIDA), which exploits the relationship of samples under various granularity. It allows us to discover multi-granularity semantic concepts as taxonomic context priors (*i.e.*, sub-class, target-class, and super-class), and then collaboratively leverage them to enhance representation learning and improve the quality of pseudo labels. Specifically, TIDA comprises two components: i) A taxonomic context discovery module that constructs a set of hierarchical prototypes in the latent space to discover the underlying taxonomic context priors; ii) A taxonomic context-based prediction alignment module that enforces consistency across hierarchical predictions to build the reliable relationship between classes among various granularity and provide additions supervision. We demonstrate that these two components are mutually beneficial for an effective OSSL framework, which is theoretically explained from the perspective of the EM algorithm. Extensive experiments on seven commonly used datasets show that TIDA can significantly improve the performance and achieve a new state of the art. The source codes are publicly available at https://github.com/rain305f/TIDA.

## 1 Introduction

Deep neural networks have obtained impressive performance on a variety of visual and language tasks [42, 48, 61, 15, 11, 12, 13, 14, 35, 36, 37, 45, 46]. However, their success is largely dependent

---

*Corresponding author.

37th Conference on Neural Information Processing Systems (NeurIPS 2023).

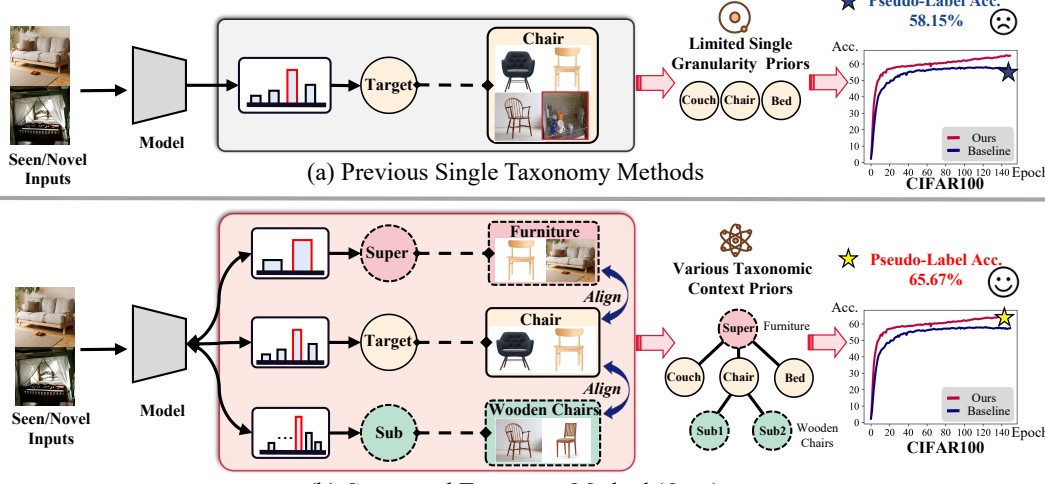

Figure 1: (a) Previous works (*e.g.* TRSSL [65]) typically focus on constraining models on the pre-defined single granularity, resulting in inferior pseudo labels. (b) Our method explicitly explores the multi-granularity priors for model training, producing more accurate pseudo labels.

on abundant labeled data, where the acquisition is expensive and time-consuming [75]. One popular solution to this problem is Semi-Supervised Learning (SSL) [69, 34, 63, 54, 62, 44, 56], which aims to reduce annotation costs by supplementing a small amount of labeled training data with a large number of unlabeled data.

Despite their significant success in various tasks [68, 77, 21, 75, 69, 53, 78, 38, 73, 59] , most existing SSL methods are designed under the closed-set assumption, *i.e.*, the training and test data share the same class set. Recently, several works [64, 26, 54, 31, 74] extend the standard SSL to an open-world setting, where the unlabeled set contains samples from novel (unknown) classes that are not present in the labeled set. In Open-world SSL (OSSL), the model is required to identify samples from both known and novel classes with partial labeled sample from seen classes. Due to the lack of labeled samples for novel classes, it is vital to exploit some priors as auxiliary supervision to learn discriminative representations and classifiers for all classes. Existing works either exploit pairwise similarity prior [64, 5, 25] or class distribution prior [65] to achieve this. However, as shown in Fig. 1(a), these methods only explore priors at a single granularity, which suffer from sub-optimization and inaccurate pseudo labels, due to the limited supervision.

In this paper, we argue that leveraging multiple levels of granularity as semantic priors (*e.g.*, subclasses, classes, and super-classes, etc.) is a more preferable solution for OSSL [43], which helps us utilize more underlying relationship to supplement simplex single supervision. As illustrated in Fig. 1(b), in the super-class granularity, we can treat *Couch* (seen) and *Chair* (novel) as belonging to the same super-class *Furniture*. This builds a relationship between seen and novel classes and constraints samples from *Chair* and *Couch* be closer [43]. Meanwhile, in the sub-class granularity, *Chairs* are over-clustered into different sub-classes according to the materials, helping us to distinguish some hard negative samples [24]. These motivate us to inject such taxonomic context into OSSL for producing more discriminative representation learning and accurate pseudo-labeling. To this end, we propose a novel OSSL framework, named **T**axonomic context pr**I**ors **D**iscovering and **A**ligning (TIDA), to explore the relationship of samples (or classes) under various granularity. Without auxiliary supervision, TIDA can automatically discover multi-granularity taxonomic context priors, which are then collaboratively leveraged to improve representation and classifier learning. Meanwhile, the alignment across granularity ensures consistent taxonomic context and additional supervision.

Specifically, we develop two modules: i) Taxonomic Context Discovery (TCD) module, which discovers the underlying taxonomic context priors by constructing a set of hierarchical prototypes to cluster samples; ii) Taxonomic Context-based prediction Alignment (TCA) module, which enforces consistency across hierarchical predictions to build reliable relationships between classes at various levels of granularity and provide additional supervision. These two modules are mutually beneficial for learning discriminative representation and accurate pseudo-labeling for OSSL. As shown in

Fig. 1(b), our method can largely improve the quality of the pseudo label over the baseline.
To sum up, the main contributions are as follows:

- We identify the importance of multi-granularity priors in the context of OSSL and introduce a new type of prior knowledge, *i.e.*, taxonomic context priors, for solving the OSSL problem.
- We introduce a uniformed OSSL framework, which can discover taxonomic context priors without any extra supervision. With the proposed cross-hierarchical prediction alignment, our framework can effectively enhance the performance of the model.
- Experiments conducted on seven visual benchmarks show that our TIDA achieves new state-of-the-art results. Additionally, we provide a theoretical analysis with the EM algorithm to better understand the underlying mechanism of our approach.

## 2  Related work

**Open-world Semi-Supervised Learning (OSSL).** Semi-Supervised Learning (SSL) aims to learn informative semantics from unlabeled data to reduce the dependence on human annotations [10, 2, 1, 39, 7, 60, 33, 3, 9, 30, 49, 67, 16, 6, 76, 4]. However, these methods assume that the unlabeled and labeled samples come from the same distribution, which is impractical in real-world applications. To address this limitation, ORCA [5] extends SSL to the more realistic and challenging open world, which assumes the unlabeled data are from both novel and labeled (seen) classes. NCLPS [25] proposes to exploit pairwise similarity priors with distribution alignment and applies an adaptive threshold to synchronize the learning pace between seen/novel classes. OpenLDN [64] optimizes a pairwise similarity loss by a bi-level way, which exploits the information available in the labeled set as priors to implicitly cluster samples from novel class. Later, TRSSL [65] proposes a class-distribution-aware pseudo-label method for OSSL, which utilizes sample uncertainty and class distribution as priors to generate pseudo-labels. However, they still only focus on priors learned on single label-hierarchy, which lacks accurate modeling of multi-level semantic relations between seen/novel category samples. For a new perspective, TIDA applies a more preferable solution, which subdivide semantic concepts from coarse to fine as taxonomic context (*i.e.*, sub-class, target-class, and super-class). With taxonomic context to describe seen/novel samples, it enjoys a number of desirable properties, including flexible encoding of label relations, predictions consistent with label relations [19]. Our model can significantly improve performance by exploiting the label relations.

**Novel Class Discovery (NCD).** NCD is a challenging open-world problem, which most closely relates to OSSL and aims at clustering unlabeled samples of novel classes with the guidance of knowledge from seen categories [70, 32, 80, 79, 50, 66, 51, 57]. RankStats [28] utilizes rank statistics to transfer the knowledge of the labeled classes to the problem of clustering the unlabelled images. UNO [23] first formulates the NCD problem into a transportation problem by Shinkhorn-Knopp algorithm [17]. Different from OSSL, NCD has a strong assumption that all the unlabeled samples belong to novel classes, thus NCD methods are not applicable to OSSL problems. In this paper, we adopt some NCD methods as comparisons, but the results show that they fail to jointly distinguish seen/novel samples.

## 3  Methodology

**Problem Setup.** Similar to standard closed-world SSL, the training data for OSSL consist of labeled data $\mathcal{D}^l = \{x_i^l, y_i^l\}_{i=1}^{N_l} \in \mathcal{X} \times \mathcal{Y}^l$ and unlabeled data $\mathcal{D}^u = \{x_i^u, y_i^u\}_{i=1}^{N_u} \in \mathcal{X} \times \mathcal{Y}^u$, where $x_i^l$ is the $i$-th labeled image with the label $y_i^l$, $x_i^u$ is the $i$-th unlabeled image belonging to the class $y_i^u$ that is not available during training, and $N_l < N_u$. Different from standard SSL, OSSL assumes that $\mathcal{Y}^l \subset \mathcal{Y}^u$. We aims to train encoder and a $|\mathcal{Y}^u|$-ways classifiers to classify all classes. We denote $\mathcal{Y}^l$ as seen classes and $\mathcal{Y}^u \setminus \mathcal{Y}^l$ as novel classes $\mathcal{Y}^n$, *i.e.*, $\mathcal{Y}^n = \mathcal{Y}^u \setminus \mathcal{Y}^l$. Following [5], we assume the number of novel classes $|\mathcal{Y}^n|$ is known.

**Baseline.** Different from standard SSL, the pseudo-labeling process is inherently ill-posed in OSSL since there are no labeled samples for novel classes. To solve this problem, TRSSL [65] and UNO [23] design a unified self-labeling classification objective based on Shinkhorn-Knopp algorithm [17]. They view pseudo-labeling as an optimal transport problem that can generate reliable pseudo-labels for unlabeled data (more details refers to [65, 23]. The objective can be formulated as:

$$\mathcal{L}_{ce} = -\sum_{j=1}^{|\mathcal{Y}^l|+|\mathcal{Y}^u|} y_i^j \log \hat{y}_i^j, \quad \hat{y}_i^j = p(j; x_i) = \frac{\exp(\mathcal{S}(z_i, c_j))}{\sum_{m=1}^{|\mathcal{Y}^l|+|\mathcal{Y}^u|} \exp(\mathcal{S}(z_i, c_m))}, \quad (1)$$

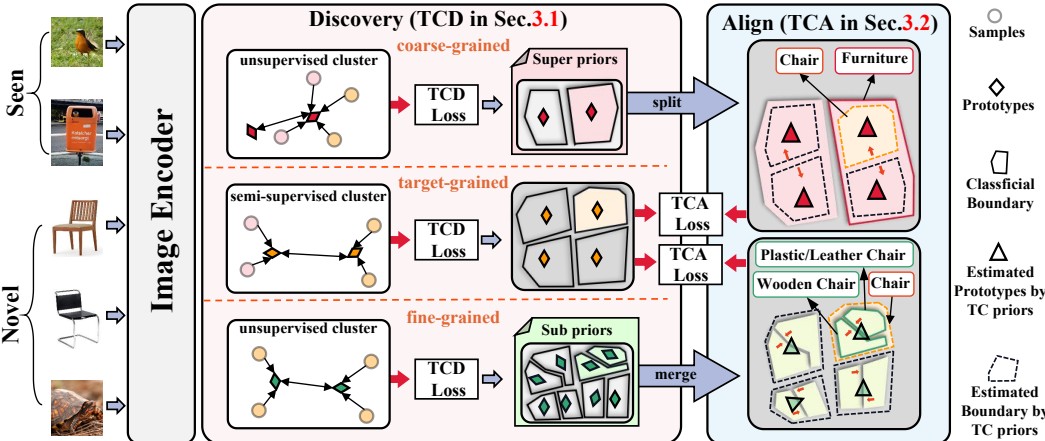

Figure 2: The overall architecture of TIDA. First, we extract features using an image encoder. Then, we use the proposed Taxonomic Context Discovery (TCD) to build hierarchical prototypes to cluster features under various granularity by optimizing TCD losses $\mathcal{L}_{tcd}$. Given the discovered taxonomic context (*i.e.* hierarchical prototypes), the proposed Taxonomic Context-based prediction Alignment (TCA) estimates taxonomic context-based predictions on target hierarchy by splitting/merging cluster results on coarse/fine-grained into the target. Last, TCA constrains the estimated predictions to be consistent with the original one on the target hierarchy with TCA losses $\mathcal{L}_{tca}$. During inference, the encoder and target-grained prototypes are applied to classify seen/novel class samples. Colors indicate different classes for samples or different granularity for prototypes.

where $y_i \in \mathbb{R}^{|\mathcal{Y}^u|+|\mathcal{Y}^l|}$ denotes the ground-truth / pseudo-label for labeled / unlabeled sample $x_i$. $\hat{y}_i^j$ is the output predicted by the model, representing the probability of $x_i$ belongs to $j$-th class. $z_i$ represents the features of sample $x_i$ and $c_j$ represents the prototype in $j$-th class. $\mathcal{S}(\cdot, \cdot)$ is cosine similarity function. Note that to mitigate over-fitting on noisy pseudo-labels, both methods first obtain pseudo labels for two views of each unlabeled sample by applying various perturbations. Then, the classification objective in Eq. 1 for unlabeled data is accomplished by optimizing each view using the pseudo-labels from the other view. For simplify, we do not explicitly display this pseudo-label exchange process in Eq. 1. In this paper, we use this self-labeling strategy as the baseline. Differently, we propose to explore underlying taxonomic context priors as a remedy for single-granularity supervision. As shown in Fig. 2, TIDA consists of two mutually benefited modules: Taxonomic Context Discovery (TCD) and Taxonomic Context-based prediction Alignment (TCA). Next, we describe TIDA in detail.

## 3.1 Taxonomic Context Discovery

The lack of supervision for novel categories in OSSL poses a challenge for models to learn accurate and discriminative features for classification. Inspired by [58, 27, 43], we argue that the underlying taxonomic context of data facilitate novel class discovery, which can provide rich relationships between samples under different category granularity for supervision. To achieve this goal, we build several learnable hierarchical prototypes $C = \{\{c_i^l\}_{i=1}^{n_l}\}_{l=1}^{L}$ upon a shared feature encoder $f_\theta : \mathcal{X} \to \mathcal{Z}$, which are normalized and can be regarded as semantic priors under various category granularity. $L$ is the number of hierarchies and $n_l$ is the number of prototypes of the $l$-th layer and $n_1 < n_2 < ... < n_L$. Note that the number of prototypes $n_l$ controls the granularity of classification, where larger/smaller $n_l$ means more fine-/coarse-grained semantic concepts at hierarchy $l$.

Specifically, TIDA first obtains normalized feature representation $z_i \in \mathbb{R}^K = f_\theta(x_i)$ for the $i$-th sample $x_i$ by image encoder $f_\theta$. Then, we use $z_i$ and prototypes on each hierarchy to perform clustering based on the baseline. The combination of loss functions on each hierarchy is written as:

$$\mathcal{L}_{tcd} = -\sum_{l=1}^{L}\sum_{j=1}^{n_l} y_i^{j,l} \log \hat{y}_i^{j,l} = -\sum_{l=1}^{L}\sum_{j=1}^{n_l} y_i^{j,l} \log \frac{\exp(\mathcal{S}(z_i, c_j^l)/\tau)}{\sum_{m=1}^{n_l} \exp(\mathcal{S}(z_i, c_m^l)/\tau)}, \quad (2)$$

where $y_i^l \in \{0,1\}^{|\mathcal{Y}^l+\mathcal{Y}^u|}$ is the pseudo-label of $x_i$ on $l$-th hierarchy generated by Sinkhorn-Knopp algorithm [17], $\hat{y}_i^l$ denotes the prediction of model for $x_i$ and $\tau = 0.1$ is the temperature. In this

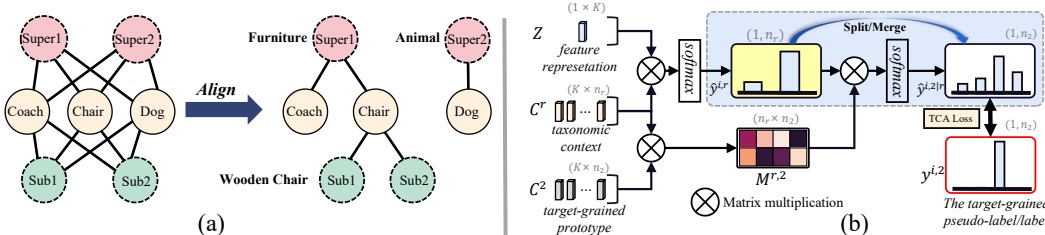

Figure 3: (a) The left part denotes inconsistent taxonomic context, where the relationships between labels across different hierarchies are unclear and inaccurate. TCA can address this problem by explicitly building affinity relationships and aligning hierarchical predictions, resulting in a consistent taxonomic context. (b) The illustration of TCA. Given the taxonomic context and the target-grained prototype, we first calculate their affinity matrix. Then, we split/merged the prediction on other-grained into target-grained as Eq. 4. Finally, we apply the loss $\mathcal{L}_{tca}$ constrains them to be consistent.

paper, we apply TIDA with three prototype hierarchies, *i.e.*, $C = \{\{c_i^1\}_{i=1}^{n_1}, \{c_i^2\}_{i=1}^{n_2}, \{c_i^3\}_{i=1}^{n_3}\}$, $n_1 = \alpha * n_2, n_2 = |\mathcal{Y}^l| + |\mathcal{Y}^u|, n_3 = \beta * |C^2|$ and $0 < \alpha < 1, \beta > 1$, which can achieve consistent well performance on all settings. More analyses about $L$ are provided in the Appendix.

**Discussion.** In fact, the objective of Eq. 2 is a hierarchical classification task, where the model requires distinguishing samples at different granularity. As shown in Fig. 2, the first hierarchy can be regarded as the super-class or coarse-grained class level; the second hierarchy refers to the target-class / target-grained level (i.e., $|\mathcal{Y}^l + \mathcal{Y}^u|$-ways classification); and the third can be regarded as the sub-class or fine-grained class level. We call the learned hierarchical clusters as taxonomic context priors because they reflect the inter-class or intra-class relationships among various granularity. However, with unknown class and limited labeled data, training each hierarchy individually cannot guarantee the **taxonomic context consistency** among hierarchies (see left part of Fig. 3(a)). That is, the relationships between labels among hierarchies are unclear and inaccurate, leading to limited advantages to target-classifier learning. With the consistent taxonomic context priors, we can build a clear subsumption relationship among classes of adjacent hierarchies and thus infer the predictions of a sample on a certain hierarchy based on its predictions on other hierarchies. For example, as shown in the right part of Fig. 3(a), given the affinities between the classes of different hierarchies, if we know the coarse-grained label of a sample is *Furniture*, we can infer that its target-grained label is most likely *Chairs* or *Coach* and definitely not *Dog*. Similarly, if we know the fine-grained label of a sample is *Wooden Chairs*, we can infer that its target-grained label is most likely *Chairs* and definitely not *Coach* or *Dog*. This indicates that the key to **taxonomic context consistency** is having reliable affinity relationships between labels of various granularity.

## 3.2 Taxonomic Context-based Prediction Alignment

To achieve taxonomic context consistency, we propose the Taxonomic Context-based prediction Alignment (TCA), as shown in Fig. 3(b), which aims to establish reliable affinity relationships across hierarchies. TCA is inspired by the statement of [19, 8]: given a sample and its label on a certain hierarchy $l$, **we can treat its unobserved labels on adjacent hierarchies as latent variables and use them to infer the probability belonging to each class on the hierarchy $l$.** Then, the tree hierarchy constraint can be achieved by minimizing the cross-entropy loss between the label/pseudo-label and the inferred prediction of the sample. To this end, we regard coarse/fine-grained predictions as latent variables and use them to infer target-grained predictions. This allows us to explicitly build affinity relationships across hierarchies and supervise them with partially labeled samples from seen classes. In addition to ensuring taxonomic context consistency, TCA also offers two more benefits: i) coarse-/fine-grained classifications can aid help predict target-grained classes [19, 8, 43]; ii) the supervision of labeled samples of seen classes can improve discrimination on coarse/fine-grained.

Specifically, given coarse/fine-grained prototypes $C^r$ and target-grained prototypes $C^2$, we first obtain the affinity matrix $M^{r,2} \in \mathbb{R}^{n_r \times n_2}$, where for each $m_{i,j}^{r,2} \in M^{r,2}$, $m_{i,j}^{r,2} = \mathcal{S}(c_i^r, c_j^2)$, $c_i^r$ is the $i$-th prototype in $C^r$, $c_j^2$ is the $j$-th one in $C^2$, and $r = 1, 3$. Second, we estimate the taxonomic context-based similarity between the feature of each sample and the target-grained prototypes $C^2$

with $C^r$ as latent variables, as follows:

$$\mathcal{S}(z_i, c_j^2) = \sum_{k=1}^{n_r} \mathcal{S}(z_i, c_k^r) \cdot m_{k,j}^{r,2} = \sum_{k=1}^{n_r} \mathcal{S}(z_i, c_k^r) \cdot \mathcal{S}(c_k^r, c_j^2). \tag{3}$$

Then, we can obtain the taxonomic context-based prediction at the target-grained level by:

$$\hat{y}_i^{j,2|r} = \frac{\exp(\sum_{k=1}^{n_r} \mathcal{S}(z_i, c_k^r)/\tau \cdot \mathcal{S}(c_k^r, c_j^2))}{\sum_{m=1}^{C^l+C^u} \exp(\sum_{k=1}^{n_r} \mathcal{S}(z_i, c_k^r)/\tau \cdot \mathcal{S}(c_k^r, c_m^2))}, \tag{4}$$

where $r$ indicates the hierarchy of $C^r$. Finally, we can enforce the taxonomic context consistency/alignment learning based on the cross-entropy loss and the pseudo-labels (or ground truth) on the target level, formulated as:

$$\mathcal{L}_{tca} = \sum_{r=1,3} \sum_{j=1}^{|\mathcal{Y}^l|+|\mathcal{Y}^u|} y_i^{j,2} \log \hat{y}_i^{j,2|r}. \tag{5}$$

Finally, we jointly perform taxonomic context discovery and taxonomic context-based prediction alignment during the training process, *i.e.*, optimizing the model with the objective of $\mathcal{L}_{tcd} + \mathcal{L}_{tca}$.

## 4   Theoretical Analysis

Assuming that the observed data $\mathcal{D}^l \cup \mathcal{D}^u$ are related to some latent variables sets which refer to the hierarchical prototypes $C = \{\{c_i^l\}_{i=1}^{n_l}\}_{l=1}^L$ above, where $n_l$ is the number of latent variables in set $C^l$, $l = 1, \cdots, L$ and $n_1 < n_2 < \cdots < n_L$. In our TIDA, $L = 3$, and $n_2 = |\mathcal{Y}^l| + |\mathcal{Y}^u|$.
From the EM [18] perspective, TIDA aims to maximize the likelihood of the observed $N_l + N_u$ samples based on these hierarchical prototypes $C$. According to Jensen's inequality [47], we write the surrogate objective function as follows:

$$\theta^*, C^* = \arg\max_{\theta, C} \sum_i^{N_l+N_u} \sum_{C^l \in C} \sum_{c_j^l \in C^l} Q(c_j^l) \log p(x_i, c_j^l; \theta) \quad \text{with} \quad Q(c_i^l) = p(c_j^l; x_i, \theta). \tag{6}$$

To sum up, TIDA aims to estimate the posterior class probability $Q(c_j^l) = p(c_j^l; x_i, \theta)$ on each hierarchy at the E-steps. Then, TIDA draws each sample to the prototype of its assigned cluster on each hierarchy at the M-step by optimizing Eq. 6 with known $Q(c_j^l)$.

**Theorem 1.** *By executing EM algorithm iteratively, samples from the same class in the feature space will be mapped into a $d$-variate von Mises-Fisher (vMF) distribution whose probabilistic density is given by $g(\boldsymbol{x}|\boldsymbol{c}_i^l, \kappa; \theta) = c_d(\kappa)e^{\kappa \boldsymbol{c}_i^{l\top} f_\theta(\boldsymbol{x})}$, where $|\boldsymbol{c}_i^l| = 1$ and $\boldsymbol{c}_i^l$ represents the mean direction, $\kappa = 1/\tau$ is the concentration parameter, and $c_d(\kappa)$ is the normalization factor.*

**E-step.** In fact, the posterior class probability $Q(c_i^l) = p(c_i^l; x_i, \theta) = \mathbb{I}(x_i \in c_j^l)$, which is equivalent to $y_i^l$ in Eq. 2. In this paper, TIDA uses the Sinkhorn-Knopp algorithm on each hierarchy to assign the samples to the cluster center, as mentioned in Sec. 3.
**M-step.** Given $Q(c_j^l) = p(c_j^l; x_i, \theta)$, TIDA aims to maximize the surrogate function in Eq. 6 in M-step. Finally, as Theorem. 1 shown, samples will be mapped into different $d$-variate von Mises-Fisher (vMF) distributions under different hierarchies [72].

*(a) Separate Objective Functions for Each Hierarchy.* Due to Eq. 6 is hard to optimize, following [47], we assume that the distribution around each prototype $c_i^l$ satisfy an isotropic Gaussian, *i.e.*, $p(x_i; c_i^l, \theta) = \exp\left(-(z_i - c_i^l)^2/2\sigma_i^{l2}\right) / \sum_{j=1}^{n_l} \exp\left(-(z_i - c_j^l)^2/2\sigma_j^{l2}\right)$. Then Eq. 6 is written as:

$$\theta^*, C^* = \arg\max_{\theta, C^l} \sum_i^{N_l+N_u} \sum_{C^l \in C} \sum_{c_i^l \in C^l} -\log \frac{\exp\left(z_i \cdot c_s^{i,l}/\tau\right)}{\sum_{j=1}^{n_l} \exp\left(z_i \cdot c_j^l/\tau\right)}, \tag{7}$$

where $c_s^{i,l}$ denotes $z_i$'s assigned prototype ($p(c_s^{i,l}; x_i, \theta) = 1$ and $c_s^{i,l} \in C^l$) and $\tau = 1 \propto \sigma_i^{l2}$, which plays the role of temperature parameter. As we can see, Eq. 7 is equivalent to Eq. 2.

*(b) Constrain the Consistency across Hierarchies in Feature Space.* Since only optimizing Eq. 7 results in inconsistencies of these learned vMF distributions across hierarchies, we propose to obtain

Table 1: Comparison with state-of-the-art methods on generic datasets.

| Methods | CIFAR10 | | | CIFAR100 | | | ImageNet-100 | | | Tiny ImageNet | | |
|---|---|---|---|---|---|---|---|---|---|---|---|---|
| | Seen | Novel | All | Seen | Novel | All | Seen | Novel | All | Seen | Novel | All |
| DTC [29] | 42.7 | 31.8 | 32.4 | 22.1 | 10.5 | 13.7 | 24.5 | 17.8 | 19.3 | 13.5 | 12.7 | 11.5 |
| RankStats [28] | 71.4 | 63.9 | 66.7 | 20.4 | 16.7 | 17.8 | 41.2 | 26.8 | 37.4 | 9.6 | 8.9 | 6.4 |
| UNO [23] | 86.5 | 71.2 | 78.9 | 53.7 | 33.6 | 42.7 | 66.0 | 42.2 | 53.3 | 28.4 | 14.4 | 20.4 |
| ORCA [5] | 82.8 | 85.5 | 84.1 | 52.5 | 31.8 | 38.6 | 83.9 | 60.5 | 69.7 | – | – | – |
| OpenNCD [50] | 83.5 | 86.7 | 85.3 | 53.6 | 33.0 | 41.2 | 84.0 | 65.8 | 73.2 | – | – | – |
| TRSSL [65] | 94.9 | 89.6 | 92.2 | 68.5 | 52.1 | 60.3 | 82.6 | 67.8 | 75.4 | 39.5 | 20.5 | 30.3 |
| OpenLDN [64] | 92.4 | 93.2 | 92.8 | 55.0 | 40.0 | 47.7 | – | – | – | – | – | – |
| TIDA (Ours) | **94.2** | **93.4** | **93.8** | **73.3** | **56.6** | **65.3** | **83.4** | **71.2** | **77.6** | **45.7** | **28.4** | **37.2** |

Table 2: Comparison with state-of-the-art methods on fine-grained datasets.

| Methods | Oxford-IIIT Pet | | | FGVC Aircraft | | | Stanford-Cars | | |
|---|---|---|---|---|---|---|---|---|---|
| | Seen | Novel | All | Seen | Novel | All | Seen | Novel | All |
| DTC [29] | 20.7 | 16.0 | 13.5 | 16.3 | 16.5 | 11.8 | 12.3 | 10.0 | 7.7 |
| RankStats [28] | 12.6 | 11.9 | 11.1 | 13.4 | 13.6 | 11.1 | 10.4 | 9.1 | 6.6 |
| UNO [23] | 49.8 | 22.7 | 34.9 | 44.4 | 24.7 | 31.8 | 49.0 | 15.7 | 30.7 |
| TRSSL [65] | 70.9 | 36.1 | 53.9 | 69.5 | 41.2 | 55.4 | 83.5 | 37.1 | 60.4 |
| OpenLDN [64] | 66.8 | 33.1 | 50.4 | – | – | 45.7 | – | – | 38.7 |
| TIDA (Ours) | **75.7** | **39.2** | **59.9** | **71.1** | **43.7** | **57.4** | **85.9** | **43.5** | **66.0** |

the aggregated latent variables $\tilde{C}^{2|r}$ with variables set $C^r$, then constrain the distribution around $\tilde{C}^{2|r}$ be consistent with the original one around $C^2$, where $r = 1, 3$.

First, we establish the similarity relationship across adjacent hierarchy, *i.e.*, the mapping matrix $M^{r,2} = \mathcal{S}(C^{r\top}, C^2)$, where for $m^{r,2}[j,k] \in M^{r,2}$, $m^{2,r}[j,k] = \mathcal{S}(c_j^r, c_k^2)$. Then we obtained its aggregated variables set $\tilde{C}^{2|r}$ with latent variables $C^r$ as $\tilde{C}^{2|r} = \{\tilde{c}_k^{2|r}\}_{k=1}^{n_2}$, $\tilde{c}_k^{2|r} = \sum_{j=1}^{n_2} c_j^r m^{r,2}[j,k]$. Note, the variables set $\tilde{C}^{2|r}$ dynamically aggregates semantics from $r$-th hierarchy and builds communication among clusters across hierarchies, which facilitates aligning hierarchical semantics with target tasks.

Given aggregated variables, we then apply consistent constraints on the sample's distribution around $\tilde{C}^{2|r}$ and the original ones in the target hierarchy, where $r = 1, 3$. Specifically, we use aggregated variables $\tilde{C}^{2|r}$ to calculate the objective function in Eq. 7, as follows:

$$\theta^*, C^* = \arg\max_{\theta, C} \sum_{i}^{N_l+N_u} \sum_{r=1,3} -log \frac{\exp\left(z_i \cdot \tilde{c}_s^{i,2|r}/\tau\right)}{\sum_{k=1}^{n_2} \exp\left(z_i \cdot \tilde{c}_k^{2|r}/\tau\right)}, \tag{8}$$

where the index $s$ in $\tilde{c}_s^{i,2|r}$ corresponds to the assigned target-grained prototype $c_s^{i,2}$ for feature $z_i$ ($p(c_s^2; x_i, \theta) = 1$). As we can see, Eq. 8 is equivalent to Eq. 5. More details are in Appendix.

## 5 Experiment

### 5.1 Experiment Setup

**Datasets.** We evaluate TIDA on four commonly used generic image classification datasets (i.e. CIFAR10 [41], CIFAR100 [41], TinyImageNet [20] and ImageNet-100 [20]) and three fine-grained datasets (i.e. Oxford-IIT Pet [52], Standford-Cars [40] and Aircraft [55]). Following [5, 65], we use the first half of classes as seen classes and the remaining as novel. More details are in the Appendix.

**Implementation Details.** Following [65, 5, 64], we use ResNet-50 [22] for ImageNet-100 and ResNet-18 [22] for the other datasets. For all datasets, we set the length of prototypes set $L$ as 3, in which $C = \{C^1, C^2, C^3\}$ and $|C^2| = |\mathcal{Y}^l \cup \mathcal{Y}^u|$. For the number of prototypes $C^1$ and $C^3$, we set $|C^1| = \alpha * |C^2|$ and $|C^3| = \beta * |C^2|$. We set $\alpha = 0.2 / 0.4$ and $\beta = 2 / 2.5$ for generic / fine-grained datasets. We use a cosine annealing-based learning rate scheduler accompanied by a linear warmup, where we set the base learning rate to 0.5 / 1.5 for generic / fine-grained datasets. The warmup length is set to 10 epochs and the weight decay is set to 1e-4. More details are in the Appendix.

Table 3: The ablation study. **C-TCP**: **C**oarse-grained **T**axonomic **C**ontext **P**riors; **F-TCP**: **F**ine-grained **T**axonomic **C**ontext **P**riors; **TCA**: **T**axonomic **C**ontext-based prediction **A**lignment. When using TCA only, the model is equipped with three target-grained classifiers that are aligned by TCA.

| # | C-TCP | F-TCP | TCA | CIFAR100 | | | Tiny ImageNet | | | Stanford-Cars | | |
|---|---|---|---|---|---|---|---|---|---|---|---|---|
| | | | | Seen | Novel | All | Seen | Novel | All | Seen | Novel | All |
| a) | | | | 67.0 | 48.9 | 57.9 | 39.7 | 21.8 | 31.1 | 84.8 | 38.3 | 61.3 |
| b) | ✔ | | | 63.7 | 47.0 | 55.5 | 36.4 | 19.2 | 28.0 | 81.2 | 41.2 | 61.1 |
| c) | ✔ | | ✔ | 71.3 | 51.6 | 61.6 | 43.7 | 27.1 | 35.8 | 84.9 | 40.8 | 62.9 |
| d) | | ✔ | | 65.3 | 45.6 | 55.3 | 36.3 | 20.4 | 28.8 | 78.1 | 31.2 | 54.8 |
| e) | | ✔ | ✔ | 71.5 | 54.6 | 63.1 | 44.6 | 27.3 | 36.8 | 85.3 | 42.3 | 64.1 |
| f) | | | ✔ | 71.8 | 49.1 | 60.5 | 43.1 | 21.1 | 32.5 | 82.8 | 39.5 | 61.9 |
| g) | ✔ | ✔ | | 69.9 | 45.4 | 57.7 | 36.3 | 19.8 | 29.0 | 76.4 | 30.2 | 53.8 |
| h) | ✔ | ✔ | ✔ | **73.3** | **56.6** | **65.3** | **45.7** | **28.4** | **37.2** | **85.9** | **43.5** | **66.0** |

## 5.2 Comparison with State-of-The-Art

We first compare our TIDA with state-of-the-art methods [5, 23, 28, 29, 64, 65] on four generic datasets and three fine-grained datasets, where the results are reported in Tab. 1 and Tab. 2, respectively. It is clear that TIDA consistently outperforms all state-of-the-art methods across all datasets and metrics. Specifically, TIDA surpasses the current best competitor TRSSL [65] by 6.9% on TinyImageNet for All classes. Importantly, TIDA outperforms previous methods on most datasets for Novel classes by a large margin, *e.g.*, 7.9% on TinyImageNet and 6.4% on Stanford-Cars. These results experimentally demonstrate that exploring taxonomic context priors as auxiliary supervision is a beneficial way for discriminating seen and novel classes under OSSL.

## 5.3 Ablation Study

**Effect of Coarse-grained Taxonomic Context Priors.** To verify the effectiveness of Coarse-grained Taxonomic Context Priors (**C-TCP**), we conduct a comparison with three variants: a) the baseline that only clusters samples on target-grained; b) the model that clusters samples on both coarse-grained and target-grained without using Taxonomic Context-based prediction Alignment (**TCA**); c) the model which simultaneously clusters samples both on coarse-grained and target-grained with TCA. As we can see in Tab. 3, c) outperforms a) and b) but b) worsened a), showing that generic semantics will improve the performance only when the coarse-grained clustering is consistent with the target task.

**Effect of Fine-Grained Taxonomic Context.** Similarly, we also conduct a comparison with three variants to verify the effectiveness of Fine-grained Taxonomic Context Priors (**F-TCP**), *i.e.*, a) the baseline, d) the model additionally includes F-TCP learning, and e) the model additionally includes F-TCP learning and TCA. Results show that F-TCP learning facilitates the target task only when using ATC. Moreover, the fine-grained semantic works relatively better than the generic semantic (see the comparison between c) and e)), especially the performance on Novel classes. In addition, jointly considering C-TCP and F-TCP could further improve the performance, as shown in variant h).

**Effect of Taxonomic Context-based Prediction Alignment.** As discussed in the previous two ablation studies, the proposed TCA is indispensable to our TIDA. Specifically, by comparing b) v.s. c), d) v.s. e), and g) v.s. h), we can observe that only using coarse-grained taxonomic context learning or fine-grained taxonomic context learning, or both of them fail to achieve improvement over the baseline. However, after injecting the proposed taxonomic context-based prediction alignment module, the performance is significantly improved. Because the inconsistency among multi-granularity classification leads to hard-optimization problem when using TCD alone. Actually, TCD applies multiple classifiers to cluster samples under different granularity. It can be regarded as multi-task learning with a shared backbone. However, due to a lack of supervision (e.g., semantic names and attributes), when using TCD alone, the training objective among classifiers is inconsistent and the optimization on the sub- and sup-classifiers is completely unsupervised. This may have a negative impact on model optimization and consequently result in performance degradation. While, the TCA learn to align the training objectives among classifiers and enable the sub- and sup-classifiers to benefit from the knowledge of labeled data on the target classifier.

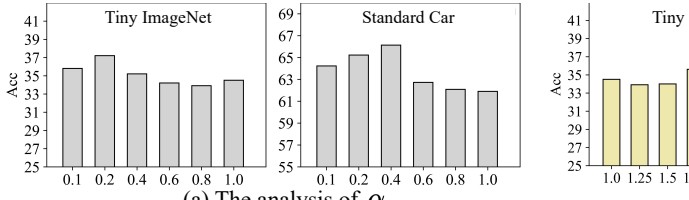

Figure 4: The hyper-parameters analysis. (a) $\alpha$: The weight that decides the number of super-classes. (b) $\beta$: The weight that decides the number of sub-classes.

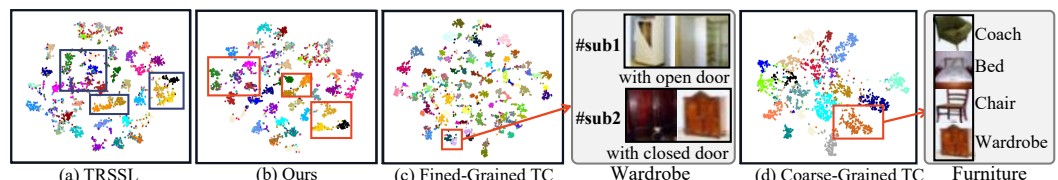

Figure 5: The t-SNE visualization of features from TRSSL and TIDA on CIFAR100. For clarity, we randomly select 20 seen and 20 novel classes as examples from CIFAR100.

**Effect of Alignment across Multiple Granularity.** It is interesting to see whether the performance improvement comes from the alignment across multiple granularity or alignment across a number of classifiers. To evaluate this, we introduce a variant f) with three target-grained clusters learning, *i.e.*, $|C^1| = |C^2| = |C^3| = |\mathcal{Y}^l| + |\mathcal{Y}^u|$. Compared with f) and h), we obverse that the alignment across multiple target-granularity is beneficial for Seen classes but there is no improvement or even reduction for Novel classes. This indicates that the improvement mainly comes from interaction among different levels of granularity instead of different classifiers of the same granularity.

## 5.4 Hyper-Parameter Analysis

**Impact of the Number of Super-Classes.** As shown in Fig. 4(a), it is more suitable to assign a smaller value for $\alpha$, which leads to a small number of coarse-grained classes. When $\alpha$ approaches to 1, TIDA degenerates to the model that ignores the coarse-grained granularity. The best result is achieved by $\alpha = 0.2$ / 0.4 for TinyImageNet / Standard Car.

**Impact of the Number of Sub-Classes.** In contrast, as shown in Fig. 4(b), assigning a larger value for $\beta$ leads to a higher performance, which enables us to obtain fine-grained prior. Again, when $\beta$ nears 1, TIDA will largely overlook the fined-grained granularity and produce lower results. The best performance is obtained by $\beta = 2$ / 2.5 for Tiny ImageNet / Standard Car.

## 5.5 Qualitative Analysis

**T-SNE Visualization.** We use t-SNE [71] to visualize the features learned by TIDA and the baseline [65]. As shown in Fig. 5(a) and Fig. 5(b), TIDA produces more discriminative features than TRSSL [65], where the samples are generally better clustered. To better understand our method, we also visualize the fine-grained and coarse-grained features learned by TIDA, shown in Fig. 5(c) and Fig. 5(d). At the fine-grained level, samples from the same categories are divided into different clusters; while, samples belonging to the same super-class are clustered together at the coarse-grained level. This further verifies that TIDA can learn more semantic priors which are complementary.

**Visualization of Affinity Matrix.** In Fig. 6, we calculate the affinity matrix by counting the number of samples belonging to each target class that are classified into each super-class. Results show that samples of the same target class are commonly classified into the same super-class for TIDA w/ TCA. In addition, we find that similar target classes are generally mapped into the same super-class. For example, "bowl, cup, lamp, plate, clock" are assigned with the 18th super-class. In contrast, for TIDA w/o TCA, samples of the same target class tend to be classified into different super-classes, resulting in unclear and inconsistent affinity relationships among classes across hierarchies. These results validate the effectiveness of our TCA in establishing a clear and consistent affinity relationship.

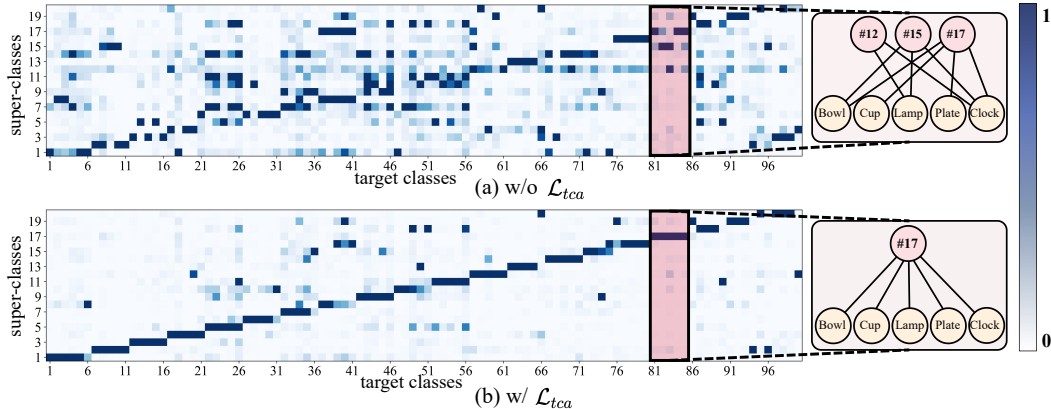

Figure 6: Affinity matrix that calculates the number of samples of each target class classified into each super-class on CIFAR100. For clarity, we rearrange the target classes, which belong to the same super-class being grouped together (see Appendix). Here, (a) and (b) follow the same order.

# 6 Contribution

In this paper, we propose a novel framework known as TIDA. It can effectively learn discriminative visual representations and improve pseudo-label accuracy by exploring the underlying taxonomic context priors. Specifically, TIDA consists of two core components: i) clustering samples hierarchically to capture the underlying taxonomic context in latent space, and ii) constraining predictions across hierarchies to be consistent to build reliable affinity relationships to ensure taxonomic context is consistent. The theoretical analysis and extensive experiments on commonly used benchmarks with consistent performance gains both justify the superiority of our TIDA.

**Limitations**. Despite obtaining high performance, TIDA and existing methods still suffer from several limitations: assuming that i) the domain distribution between labeled and unlabeled data is the same; ii) the class-distribution is uniform; and iii) the number of novel classes is known. Future work may focus on exploring more practical OSSL situations that unlock the above constraints.

**Acknowledgements.** This work was supported in part by the National Key R&D Program of China (No.2022ZD0118201), the Natural Science Foundation of China (No. 61972217, 32071459, 62176249, 62006133, 62271465), the MUR PNRR project FAIR (PE00000013) funded by the NextGenerationEU, and the EU project Ai4Media (No. 951911).

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
