# Discover and Align Taxonomic Context Priors for Open-world Semi-Supervised Learning (Supplementary Materials)

## Contents

37th Conference on Neural Information Processing Systems (NeurIPS 2023).

Table I: Statistics of the datasets and the splits for generic datasets.

| | | CIFAR10 [10] | CIFAR100 [10] | TinyImageNet [4] | ImageNet-100 [4] |
|---|---|---|---|---|---|
| Labelled | # Classes | 5 | 50 | 100 | 50 |
| | # Images | 2,500 | 2,500 | 5,000 | 6,500 |
| Unlabelled | # Classes | 10 | 100 | 200 | 100 |
| | # Images | 47,500 | 47,500 | 95,000 | 123,500 |
| Test | # Classes | 10 | 100 | 200 | 100 |
| | # Images | 10,000 | 10,000 | 100,000 | 5,000 |

Table II: Statistics of the datasets and the splits for fine-grained datasets.

| | | Oxford-IIIT Pet [12] | Aircraft [13] | Standard Car [9] |
|---|---|---|---|---|
| Labelled | # Classes | 19 | 50 | 98 |
| | # Images | 942 | 1,684 | 2,054 |
| Unlabelled | # Classes | 37 | 100 | 196 |
| | # Images | 2738 | 4,983 | 6,090 |
| Test | # Classes | 37 | 100 | 196 |
| | # Images | 3,669 | 3,333 | 8,041 |

The appendix is organized as follows: First, we provide more discussion about our method (Sec. A). Second, we provide a detailed description of the datasets and the experimental setup in Sec. B. Then, we provide additional ablation analysis and experimental results in Sec. D. Next, we provide additional quantitative analysis (Sec. E) and additional qualitative analysis about our proposed TIDA (Sec. F). Finally, we present the training algorithm of the proposed TIDA in Sec. G and its more details in Sec. H.

## A Societal Discussions

In this paper, we present a novel algorithm TIDA that significantly improves the performance of models in open-set semi-supervised learning. We summarize the potential impact of our work as follows:

**To the research community.** Our study uncovers a significant observation that incorporating taxonomic context as priors can enhance the performance of our model in challenging and meaningful real-world scenarios with limited supervision and unknown semantic concepts. This provides a new idea for utilizing unlabeled data, not only limited to open-set semi-supervised learning. Specifically, conventional approaches rarely consider the taxonomy characteristics of images, but we point out that this prior can implicitly mine the semantic structure of unlabeled data to recognize novel categories. We wish that this methodology can be generalized to more relevant label-efficient tasks.

**To label-efficient learning.** Currently, for a wide range of computer vision tasks, supervised learning with a large number of fine annotations is still the mainstream solution to achieve promising performance. However, the cost of annotation is expensive, and annotations in some scenes are difficult to obtain. Our work takes a step forward in solving this problem. With a well-trained open-set semi-supervised learning paradigm, the need for precisely annotated data would be significantly reduced, which may promote the application of AI models in annotation-difficult areas.

## B Datasets and Implementation Details

### B.1 Datasets Details

In this paper, We evaluate TIDA on four commonly used generic image classification datasets (i.e. CIFAR10 [10], CIFAR100 [10], TinyImageNet [4] and ImageNet-100 [4]) and three fine-grained datasets (i.e. Oxford-IIT Pet [12], Standford-Cars [9] and Aircraft [13]). Follow [1, 16], we use the first half of classes as seen and the remaining as novel. Since Oxford-IIIT Pet dataset contains odd number of classes, we treat the first 19 classes of this dataset as seen and the remaining 18 classes as novel [16]. The details is shown in Tab. I and Tab. II.

Table III: Comparison with state-of-the-art methods with ViT16 as the backbone.

| Methods | CIFAR10 | | | CIFAR100 | | | Aircraft | | | SCar | | |
|---|---|---|---|---|---|---|---|---|---|---|---|---|
| | Seen | Novel | All | Seen | Novel | All | Seen | Novel | All | Seen | Novel | All |
| DCCL [14] | 96.5 | 96.9 | 96.3 | 76.8 | 70.2 | 75.3 | - | - | - | 55.7 | 29.9 | 43.1 |
| PromptCAL [20] | 96.6 | **98.5** | 97.9 | **84.2** | 75.3 | 81.2 | 52.2 | **52.3** | 52.2 | 70.1 | 40.6 | 50.2 |
| GCD [17] | **97.9** | 88.2 | 91.5 | 76.2 | 66.5 | 73.0 | 41.1 | 46.9 | 45.0 | 57.6 | 29.9 | 39.0 |
| SimGCD [19] | 95.1 | 98.1 | 97.1 | 81.2 | 77.8 | 80.1 | 59.0 | 51.8 | 54.2 | 71.9 | 45.0 | 53.8 |
| PIM [2] | 97.4 | 93.3 | 94.7 | **84.2** | 66.5 | 78.3 | - | - | - | 66.9 | 31.6 | 43.1 |
| TIDA (Ours) | **97.9** | **98.5** | **98.2** | 83.8 | **80.7** | **82.3** | **61.3** | 52.1 | **54.6** | **72.3** | **46.2** | **54.7** |

**Generic Datasets.** For the generic datasets, we random use 10% datat from seen classes as labeled, the remaining 90% samples from seen classes and all samples from novel dataset as unlabeled.

**Fine-grained Datasets.** For the fine-grained datasets, we random select 50% data from seen as labeled, the remaining 50% seen samples and all novel samples as unlabeled.

## B.2 Implementation Details

**Baseline Details.** We compare our TIDA with three state-of-the-art open-world SSL methods (*e.g.*, TRSSL [16], OpenLDN [15] and ORCA [1]), three NCD methods (*e.g.*, UNO [6] and RankStats [7]), a robust SSL methods (DTC [8]). For the sake of experimental fairness, we directly adopt the results in their papers to evaluate. For methods in some related tasks (*e.g.*, NCD methods, robust SSL methods and standard SSL methods), we use the results reported in TRSSL [16].

**Experiment Setup.** Following [16, 1, 15], we use ResNet-50 [5] for ImageNet-100 and ResNet-18 [5] for the other datasets. For all experiments, we train our model for 200 epochs. We set batch size to 256 for all of our experiments except ImageNet-100 and TinyImageNet we set it to 512. We use a cosine annealing based learning rate scheduler accompanied by a linear warmup, where we set the base learning rate to 0.5 for all generic datasets while 1.0 for all fine-grained datasets, and set the warmup length to 10 epochs. We set the weight decay to 1e-4. Our experiments are conducted on NVIDIA V100 GPUs.

**Evaluation Protocol.** For evaluation, in this paper, we report the accuracy scores for seen classes $\mathcal{Y}^l$, novel classes $\mathcal{Y}^n$ and all classes $\mathcal{Y} = \mathcal{Y}^l \cup \mathcal{Y}^u$. For seen classes, we report standard classification accuracy. For novel classes and all classes, we follow [1, 16, 15] to evaluate clustering accuracy. Specifically, we consider the class prediction as cluster-ID. Next, all clusters are mapped through the optimal assignment solved by the Hungarian algorithm to their ground-truth classes.

## C Additional Comparisons

### C.1 Additional comparisons with ViT16 as the backbone

In this section, we use ViT16 as the backbone and compare our TIDA with more baselines [14, 20, 17, 19, 2]. Following [14, 20, 17, 19, 2], we incorporate contrastive learning during training. Specially, we apply self-supervised contrastive learning on all samples and apply supervised contrastive learning on labeled samples. The results are reported in Tab. III. It is clear that TIDA achieves the best performance across all datasets and metrics, except for the CIFAR100-Seen accuracy and Aircraft-Novel accuracy which are slightly lower than PromptCAL [20]. It is worth noting that PromptCAL [20] is two stages while TIDA is end-to-end, which means that our TIDA outperforms PromptCAL [20] in time-cost and computational cost. It is surprising to observe that TIDA surpasses the state-of-the-art methods by 2.9% and 1.2% on CIFAR100-Novel and Scar-Novel, which demonstrates the superiority of the taxonomic context priors in our TIDA. These results demonstrate the superiority of our method over the state-of-the-art methods.

### C.2 Additional comparisons with imbalanced dataset

Even though most standard benchmark vision datasets follows the balanced distribution, in real-world this is hardly the case. In this section, we compare TIDA with TRSSL [16] on imbalanced data to demonstrate the superiority of our TIDA. Here, we discuss two realistic imbalance scenarios: i) imbalanced class distribution and ii) imbalanced similarity in the semantic space.

Table IV: The comparison in class-imbalanced setting (CIFAR100).

| $\gamma = 10$ | Class imbalanced priors | KL | Seen | Novel | All |
|---|---|---|---|---|---|
| TRSSL [16] | ✓ | 0.0020 | 52.9 | 27.4 | 41.0 |
| TRSSL [16] | | 0.0026 | 52.3 | 23.6 | 38.9 |
| TIDA | | 0.0017 | 54.7 | 31.8 | 43.6 |

Table V: The comparison in similarity-imbalanced setting (IS-CIFAR90).

| | Seen | Novel | All |
|---|---|---|---|
| TRSSL [16] | **62.7** | 33.6 | 51.8 |
| TIDA | 62.2 | **38.7** | **53.4** |

**Imbalanced class distribution.** The learned multi-granularity pseudo-labels might be slightly biased due to our assumption of class balance. Here, we conduct experiments following the TRSSL [16]. Specifically, we use a imbalanced factor ($\gamma$) to control the distribution on CIFAR-100. We report the average accuracy on all classes and the KL diversity of pseudo-label and Gound-Truth, as shown Tab. IV. Results show that our TIDA outperforms TRSSL [16].

**Imbalanced similarity in the semantic space.** For imbalanced similarity setting, TIDA still works well and tend to learn balanced distribution. But due to the imbalanced similarity, the learned super-classes are prone to represent different grained semantics. For examples, a tiny dataset contains 6 target classes (bed, chair, couch, wardrobe, baby, dog) and the super-classes is set to be 3. The learned super-classes include #1 (furniture1: chair, couch), #2 (furniture2: wardrobe, bed), and #3 (biology: baby, dog). In other words, the granularity of semantic expressed by super class #1/2 and #3 is different.Moreover, we also verify the effectiveness of TIDA in a similarity-imbalanced setting. We randomly sample 90 classes from CIFAR100 to construct an imbalanced similarity in the semantic space (IS-CIFAR90), where the number of subclasses varies for each superclass. We report the average accuracy on Seen, Novel and ALL classes, as shown Tab. V. Results show that our TIDA surpasses TRSSL [16] by 1.6% and 5.1% on the accuracy for All and Novel classes, respectively. Meanwhile, on the accuracy for Seen classes, TIDA is only lower than TRSSL [16] by 0.5%. This indicates that, when target-classes are imbalanced in semantic distances, the learned consistent taxonomic context priors can effectively improve the recognition of novel classes classes without hampering the performance of seen classes.

Despite the improvements achieved above, the class-imbalanced assumption in our TIDA may not always be well-suited. Thus, we will further investigate these two mentioned settings in future.

## D   Additional Ablation analysis and Experiments

### D.1   Effect of the Layer Number $L$

In this section, we investigate the impact of the number of layers ($L$) for TIDA on performance, where $L$ controls the number of layer in the hierarchy prototypes $C = \{\{c_j^l\}_{i=1}^{n_l}\}_{l=1}^{L}$. To this end, we conduct a series of experiments by varying the number of layer $L$ on two generic datasets (*i.e.* CIFAR100 [10], Tiny ImageNet [4]) and two fine-grained datasets (*i.e.* Aircraft [13] and Standard Car [9]). For all experiments, we denote $n_l = \lambda_l * (|\mathcal{Y}^l| + |\mathcal{Y}^u|)$, where $\lambda_l$ control the number of prototypes in $C^l = \{c_j^l\}_{i=1}^{n_l}$ and $l = 1...L$. Here, $\lambda_1 < \lambda_2 < ... < \lambda_L$ and $l$-th layer is target-grained

Table VI: The number of prototypes on each layer in details. The table shows the number of prototypes on each layer, which used in ours experiments about the number of layer $L$ (Sec.D.1). For all experiments in Tab. VII, we denote $n_l = \lambda_l * (|\mathcal{Y}^l| + |\mathcal{Y}^u|)$, where $l = 1...L$. The red denotes additional layer compared with the setting of "$L = 3$". The "C" denotes Coarse-grained level, the "T" means Target-grained level and the "F" means Fine-grained level.

| Layer $L$ | generic datasets $n_1...n_L$ | fine-grained datasets $n_1...n_L$ |
|---|---|---|
| 3 (1C+1T+1F) | $\lambda_1 = 0.2, \lambda_2 = 1, \lambda_3 = 2$ | $\lambda_1 = 0.4, \lambda_2 = 1, \lambda_3 = 2.5$ |
| 4 (1C+1T+2F) | $\lambda_1 = 0.2, \lambda_2 = 1, \lambda_3 = 2, \lambda_4 = 3$ | $\lambda_1 = 0.4, \lambda_2 = 1, \lambda_3 = 2.5, \lambda_4 = 3$ |
| 4§ (2C+1T+1F) | $\lambda_1 = 0.2, \lambda_2 = 0.4, \lambda_3 = 1, \lambda_4 = 2$ | $\lambda_1 = 0.2, \lambda_2 = 0.4, \lambda_3 = 1, \lambda_3 = 2.5$ |
| 5 (2C+1T+2F) | $\lambda_1 = 0.2, \lambda_2 = 0.4, \lambda_3 = 1, \lambda_4 = 2, \lambda_5 = 3$ | $\lambda_1 = 0.2, \lambda_2 = 0.4, \lambda_3 = 1, \lambda_4 = 2, \lambda_5 = 3$ |

Table VII: Effect of the layer numbers $L$. The details of setting refers to Tab. VI.

| Layer $L$ | CIFAR100 | | | Tiny ImageNet | | | Aircraft | | | Standard Car | | |
|---|---|---|---|---|---|---|---|---|---|---|---|---|
| | Seen | Novel | All | Seen | Novel | All | Seen | Novel | All | Seen | Novel | All |
| 3 (1C+1T+1F) | **73.3** | **56.6** | **65.3** | **45.7** | **28.4** | **37.2** | 71.1 | 43.7 | 57.4 | 85.9 | 43.5 | 66.0 |
| 4 (1C+1T+2F) | 69.4 | 48.6 | 59.2 | 39.5 | 21.2 | 30.4 | **71.3** | 40.0 | 55.7 | 85.9 | 43.4 | 64.8 |
| 4§ (2C+1T+1F) | 69.4 | 52.7 | 61.0 | 39.4 | 24.5 | 32.4 | 70.2 | 46.2 | 58.6 | **86.0** | **44.1** | **66.5** |
| 5 (2C+1T+2F) | 72.5 | 51.4 | 62.0 | 39.3 | 20.2 | 30.0 | 71.1 | **48.5** | **59.8** | 85.2 | 40.5 | 63.0 |

layer if $\lambda_l = 1$. For different $L$, we set different $\lambda_l$ to analysis the effect of classification on each granularity for performance. Specifically, the setting about $\{n^l\}_{l=1}^L$ is shown in Tab. VI in details and the results are reported in Tab. VII. We notice that TIDA yield worse performances on generic datasets with the larger $L$, while the larger $L$ enhances performance on fine-grained datasets, *i.e.* "$L = 5$" on Aircraft and "$L = 4§$" on Standard Car. Moreover, compared with "$L = 4$", "$L = 4§$" and "$L = 3$", it is clear that the additional coarse-grained semantic ("$L = 4§$") works relatively better than the fine-grained ("$L = 4$") semantic. For simplicity and effectiveness, we set "$L = 3$".

### D.2 Ablation study about the cluster algorithm

In this paper, we adopt Sinkhorn as the online clustering algorithm due to its superiority, wide application, simplicity, and distribution-based learning strategy. Following your suggestion, we apply our approach on ORCA[25] and RankStats[24] and find that the model is hard to converge. The main reason may be that these two methods mainly reply on pair-wise similarity constraint to optimize classifier while our TIDA is constrained on the overall class distribution.

### D.3 More sensitive analysis about the hyperparameters $\alpha$ and $\beta$

In this section, we provide the justification of $\alpha$ and $\beta$ on four datasets, as shown Tab. VIII and Tab. IX. We can observe that the values of $\alpha$ and $\beta$ that yield the best performance are similar for generic datasets ($\alpha = 0.2$, $\beta = 2.0$) and fine-grained datasets ($\alpha = 0.4$, $\beta = 2.5$) respectively. We thus use one dataset to determine the values of $\alpha$ and $\beta$ for the generic (or fine-grained) setting and apply the same values of $\alpha$ and $\beta$ to all generic (or fine-grained) datasets.

Table VIII: The sensitive analtsis for $\alpha$.

| Dataset Type | Dataset | 0.1 | 0.2 | 0.4 | 0.6 | 0.8 |
|---|---|---|---|---|---|---|
| Generic | CIFAR100 | 63.4 | 65.3 | 64.7 | 61.3 | 59.3 |
| Generic | Tiny ImageNet | 35.8 | 37.2 | 35.2 | 34.2 | 33.9 |
| Fine-Grained | Standard Cars | 64.2 | 65.2 | 66.1 | 62.7 | 62.1 |
| Fine-Grained | Oxford-IIIT Pets | 58.4 | 59.4 | 59.9 | 58.6 | 54.6 |

Table IX: The sensitive analysis for $\beta$.

| Dataset Type | Dataset | 1.25 | 1.5 | 1.75 | 2.0 | 2.5 | 3.0 |
|---|---|---|---|---|---|---|---|
| Generic | CIFAR100 | 61.9 | 62.1 | 63.6 | 65.3 | 61.7 | 61.8 |
| Generic | Tiny ImageNet | 33.9 | 34.0 | 35.6 | 37.2 | 36.7 | 34.2 |
| Fine-Grained | Standard Cars | 62.5 | 64.9 | 65.9 | 65.4 | 66.1 | 65.3 |
| Fine-Grained | Oxford-IIIT Pets | 54.1 | 55.3 | 58.3 | 60.2 | 59.9 | 59.4 |

### D.4 Experiments with the Estimated Number of Novel Classes

In this section, we verify the effective and generalization of TIDA, particularly when the number of novel classes is unknown. To estimate the number of novel classes, we carry out the k-means clustering algorithm with the varied number of cluster, following previous methods [16, 18, 6]. Then, the number of cluster $k$ with the best clustering accuracy will be used as estimated number of classes, more details refers to [16, 18, 6]. The estimated result is reported in Tab. X on several datasets. With the estimated number of novel classes, we compare the performances of both our TIDA and baseline (TRSSL [16]) on these datasets. The results are reported inn Tab. XI. As we can see, our

Table X: The estimated number of class on generic dataset.

| CIFAR10 | CIFAR100 | Tiny ImageNet | ImageNet100 |
|---------|----------|---------------|-------------|
| 10 | 117 | 139 | 192 |

Table XI: Results on generic datasets with the estimated number of novel classes.

| Methods | CIFAR10 | | | CIFAR100 | | | ImageNet-100 | | | Tiny ImageNet | | |
|---------|---------|-------|------|----------|-------|------|--------------|-------|------|---------------|-------|------|
| | Seen | Novel | All | Seen | Novel | All | Seen | Novel | All | Seen | Novel | All |
| TRSSL [16] | 94.9 | 89.6 | 92.2 | 60.7 | 44.3 | 54.1 | 82.0 | 58.6 | 70.4 | 39.1 | 21.0 | 29.1 |
| TIDA | **94.2** | **93.4** | **93.8** | **66.2** | **50.6** | **60.0** | **83.0** | **60.1** | **71.1** | **41.8** | **26.4** | **33.1** |

TIDA obtains superior performance on all datasets, which experimentally demonstrates that the generalization of our TIDA.

# E    Additional Quantitative Analysis

## E.1    The Accuracy of Pseudo Labels During Training

In this section, we compare the accuracy of pseudo-label TIDA and baseline during training on CIFAR100 and Tiny ImageNet. The results are reported in Fig. I. It is clear that with taxonomic context as priors, TIDA obtains more accurate pseudo labels during training than baseline, thus enhancing performance by a large margin.

## E.2    The Accuracy of Pseudo Labels Per Class

In this section, we compare the accuracy of pseudo-label per class TIDA and baseline on CIFAR100. The results are reported in Fig. II(a). As we can see, with taxonomic context as priors, TIDA obtains more accurate pseudo labels for each classes than baseline, thus improving performance.

## E.3    The Accuracy of Pseudo Labels Per Sub-class

In this section, we further investigate the accuracy of pseudo-label per sub-class TIDA on CIFAR100. We count the proportion of samples from different target class in each subclass and use the highest one as the pseudo-label accuracy per subclass. The pseudo-label accuracy per subclass can reflect

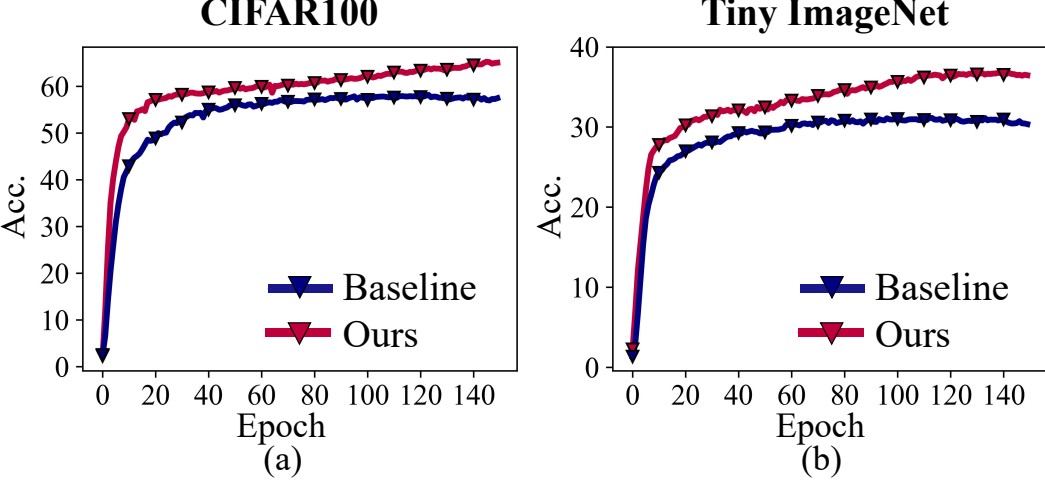

Figure I: The comparison of pseudo-label accuracy learned by TIDA and baseline. TIDA Produce more accurate pseudo-labels for target categories classification.

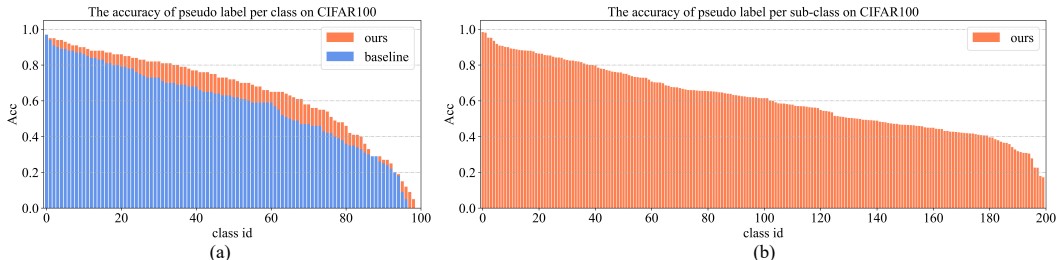

Figure II: (a) The comparison of pseudo-label accuracy per class learned by TIDA and baseline. (b) The accuracy of pseudo-label per sub-class learned by TIDA.

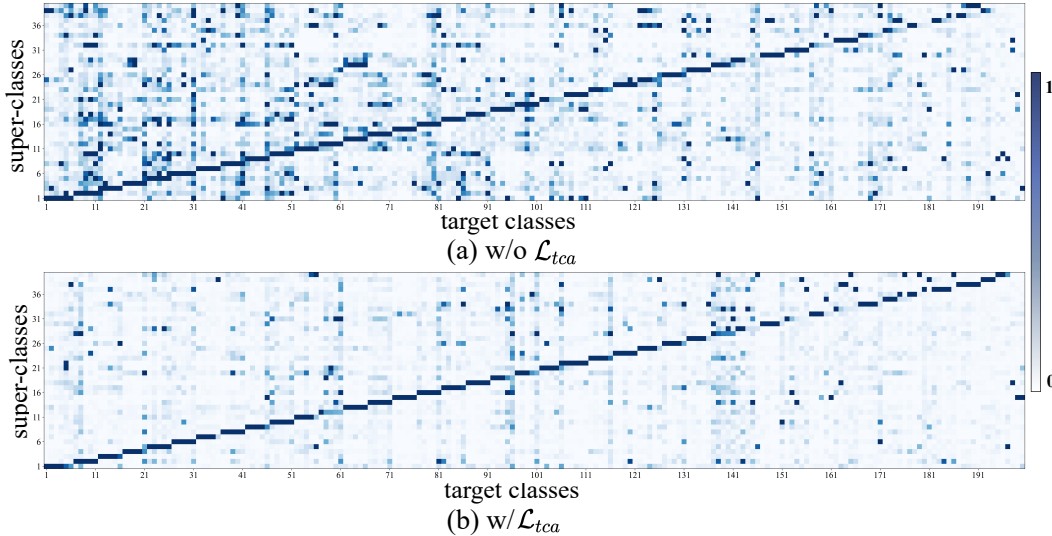

Figure III: Affinity matrix that calculates the number of samples of each target class classified into each super-class on Tiny ImageNet, the details in Sec. F.1

the performance of the subclass classification, i.e., an ideal subclass should contain only one target class sample. The result is reported in Fig. II(b), where the pseudo-label is more accurate on each sub-class than the original target classes. It provides evidence that fine-grained semantic learned by TIDA is helpful for recognising hard confused samples by over-clustering samples, thus improving performance. More visual results can be found in Sec. V.

# F   Additional Qualitative Analysis

## F.1   Visualization of Affinity Matrix

In this section, we also calculate the affinity matrix $A^{1,2} \in \mathbb{R}^{n_1 \times n_2}$ by counting the number of samples belonging to each target class that classified into each super-class on Tiny ImageNet. The item $a_{i,j}^{1,2}$ in $i$-th row and $j$-th column of $A^{1,2}$ denotes the number of samples from $j$-th target class and are classified into $i$-th super-class. We first divide by the max value of each column to normalize the affinity matrix. For clarity and comparison, we rearrange the target classes by assigning the 20% most frequently occurring target classes within each super-class to that respective super-class. This reordering strategy aims to group together target classes that share the same super-class, thereby enhancing the discernibility of class relationships. The results are reported in Fig. III. Results show that samples of the same target class are commonly classified into the same super-class for TIDA w/ TCA. In addition, we find that similar target classes are generally mapped into the same super-class. In contrast, for TIDA w/o TCA, samples of the same target class tend to be classified into different super-classes, resulting in unclear and inconsistent affinity relationships among classes across hierarchies. These results validate the effectiveness of our TCA in establishing a clear and consistent affinity relationship.

| | |
|---|---|
| 1 | 'butterfly', 'cockroach', 'orchid', 'snail', 'spider' |
| 2 | 'bed', 'can', 'chair', 'couch', 'table', 'television', 'wardrobe' |
| 3 | 'lawn_mower', 'lion', 'motorcycle', 'tiger', 'tractor' |
| 4 | 'bridge', 'forest', 'mushroom', 'road', 'skyscraper' |
| 5 | 'beaver', 'fox', 'leopard', 'porcupine', 'shrew' |
| 6 | 'elephant', 'maple_tree', 'oak_tree', 'willow_tree' |
| 7 | 'aquarium_fish', 'flatfish', 'lobster', 'ray' |
| 8 | 'bee', 'beetle', 'sunflower', 'rose', 'poppy' |
| 9 | 'bear', 'crocodile', 'otter', 'seal', 'whale' |
| 10 | 'dinosaur', 'mouse', 'palm_tree', 'pine_tree', 'turtle' |
| 11 | 'cattle',  'kangaroo', 'skunk', 'squirrel', 'wolf' |
| 12 | 'bottle', 'castle', 'keyboard', 'rocket', 'tank', 'telephone' |
| 13 | 'camel', 'cloud', 'mountain', 'plain', 'sea' |
| 14 | 'bicycle', 'caterpillar', 'crab', 'snake', 'worm' |
| 15 | 'dolphin', 'lizard', 'shark', 'trout' |
| 16 | 'baby', 'boy', 'girl', 'man', 'woman' |
| 17 | 'bowl', 'clock', 'cup', 'lamp', 'plate' |
| 18 | 'chimpanzee', 'hamster', 'possum', 'rabbit', 'raccoon' |
| 19 | 'apple', 'orange', 'pear', 'sweet_pepper', 'tulip' |
| 20 | 'bus', 'house','pickup_truck', 'streetcar', 'train' |

Figure IV: The illustration of mapping relationship between super-classes and target classes on CIFAR100. TIDA groups the 100 target-classes in the CIFAR-100 into 20 super-classes.

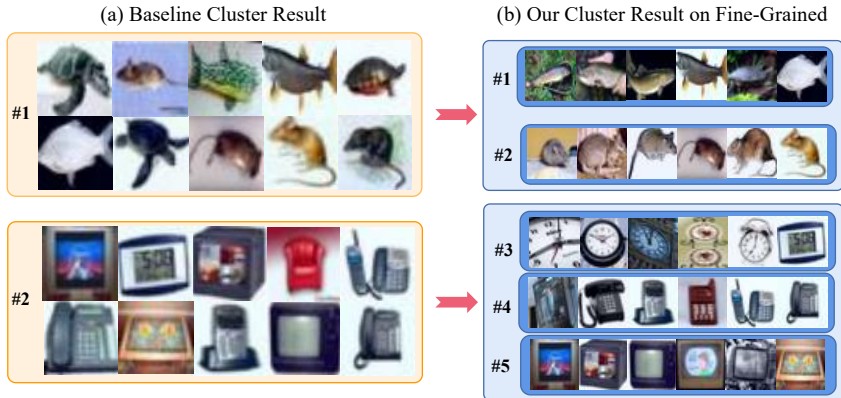

Figure V: The visualization of clustering results on fine-grained on CIFAR100. TIDA enables over-class samples on fine-grained level, which helps to recognize some hard confused-samples.

## F.2   Illustration of Mapping Relationship between Super-Classes and Target Classes

To further analyze learned semantic on coarse-grained level, we assigned each target class to the super-class where samples from this target class most likely be classified. Then we build a relationship between target classes and super-classes, as shown in Fig. IV. It is clear that the similar target classes are generally mapped into the same super-class, *e.g.*, 16-th super class ("people") contains five target classes ("baby, boy, girl, man, woman" ). The results illustrate that the proposed TIDA can indeed capture rich and helpful coarse-grained semantic for open-world semi-supervised learning (OSSL).

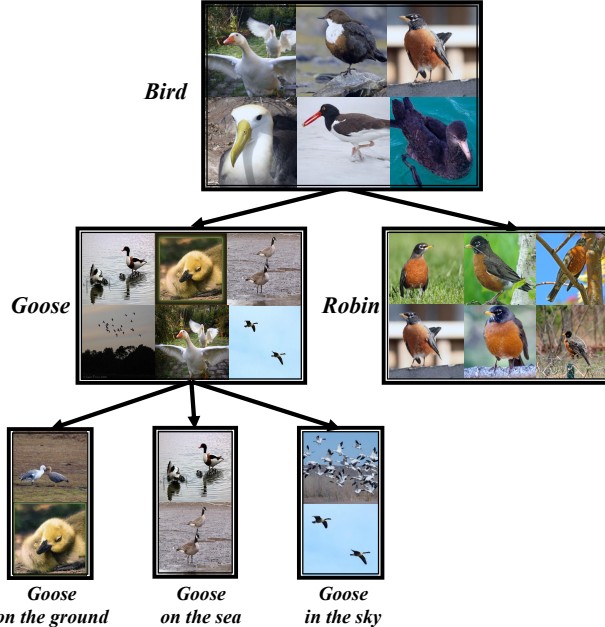

Figure VI: The visualization of hierarchical semantic structure learned by TIDA on ImageNet100.

### F.3 Visualization of Clustering Results on Fine-Grained Level

In this section, we visualize the cluster results on fined-grained level, shown in Fig. V(b). As the Fig. V(a) shown, the baseline (TRSSL [16]) tend to misclassify some hard samples on target-grained level, which leads to a sub-optimal representation and inaccurate pseudo-labels. Compared with it, our proposed TIDA enforce to distinguish these hard-to-distinguish samples by over-clustering samples, which facilitates the target task.

### F.4 Visualization of Hierarchical Semantics

We also visualize the hierarchical structure learned by TIDA, as shown in Fig. VI. The classification at top hierarchy are most diverse, which represents the coarse-grained semantics of "Bird". While, the classification at bottom hierarchy express finer-grained semantics, *e.g.*, "Goose on the ground", "Goose on the sea " and "Goose in the sky ". These results illustrate that our proposed TIDA can capture hierarchical semantic to help novel class discovery.

## G  Training Algorithm

We present our training algorithm in Alg. 1. Given labeled dataset, $\mathcal{D}^l$ and unlabeled dataset $\mathcal{D}^u$ as training data, our proposed Taxonomic context pIrors Discovering and Aligning (TIDA) aims to train an encoder $f_\theta(\cdot)$ and target-grained $C^t$ to accurately classify samples from seen/novel classes. To achieve it, we propose two module: Taxonomic Context Discovery (TCD) and Taxonomic Context-based prediction Alignment (TCA) to discovery taxonomic context as priors. First, we extract features $Z$ using an image encoder. Then, we use the proposed TCD to build hierarchical prototypes to cluster features under various granularity by optimizing TCD losses $\mathcal{L}_{tcd}$. Given the discovered taxonomic context (*i.e.* hierarchical prototypes), the proposed TCA estimates taxonomic context-based predictions on target hierarchy by splitting/merging cluster results on coarse/fine-grained into the target. Last, TCA constrains the estimated predictions to be consistent with the original one on the target hierarchy with TCA losses $\mathcal{L}_{tca}$. During inference, the encoder $f_\theta$ and target-grained prototypes $C^t$ are applied to classify seen/novel class samples. Note, we also include a MLP layer as projector on each hierarchical layer, which maps the feature $Z$ to feature spaces with different granularity. For simplify, we do not explicitly display these projector $\{g_\phi\}_{l=1}^L$.

---

**Algorithm 1** Training algorithm.

---

**Require:** Labeled dataset, $\mathcal{D}^l$, unlabeled dataset $\mathcal{D}^u$, an encoder $f_\theta(\cdot)$, the learnable hierarchical prototypes $C = \{\{c_j^l\}_{i=1}^{n_l}\}_{l=1}^{L} = \{C^l\}_{l=1}^{L}$, max iterations $K$, temperature $T$, the number of Layer $L$.

1: Initialize encoder $f_\theta(\cdot)$.
2: **for** iteration $k = 1, ..., K$ **do**
3: $\{\mathbf{X}_l, \mathbf{Y}_l\} \longleftarrow \text{MiniBatch}(\mathcal{D}^l)$
4: $\{\mathbf{X}_{u,v_1}, \mathbf{X}_{u,v_2}\} \longleftarrow \text{MiniBatch}(\mathcal{D}^u)$
5: $\mathbf{Z}_l \longleftarrow f_\theta(\mathbf{X}_l)$
6: $\mathbf{Z}_{u,v_1}, \mathbf{Z}_{u,v_2} \longleftarrow f_\theta(\mathbf{X}_{u,v_1}), f_\theta(\mathbf{X}_{u,v_2})$
7: $\mathbf{X}, \mathbf{Z} \longleftarrow \text{Concat}(\{\mathbf{X}_l, \mathbf{X}_{u,v_1}, \mathbf{X}_{u,v_2}\}), \text{Concat}(\{\mathbf{Z}_l, \mathbf{Z}_{u,v_1}, \mathbf{Z}_{u,v_2}\})$
  \*\*\*\*\*\*\*\*\*\*\*\*\*\*\*\*\*\*\*\*\*\*\*\**Taxonomic Context Discovery*\*\*\*\*\*\*\*\*\*\*\*\*\*\*\*\*\*\*\*\*
8: **for** layer $l = 1, ..., L$ **do**
9:  **if** layer $l$ is not target-grained layer $t$ **then**
10:   $\widetilde{\mathbf{Y}}_l^l \longleftarrow \text{Sinkhorn}(\mathbf{Z}_l, C^l)$
11:  **else**
12:   $\widetilde{\mathbf{Y}}_l^l \longleftarrow \mathbf{Y}_l$
13:  **end if**
14:  $\widetilde{\mathbf{Y}}_{u,v_1}^l, \widetilde{\mathbf{Y}}_{u,v_2}^l \longleftarrow \text{Sinkhorn}(\mathbf{Z}_{u,v_1}, C^l), \text{Sinkhorn}(\mathbf{Z}_{u,v_2}, C^l)$
15:  $\widetilde{\mathbf{Y}}^l \longleftarrow \text{Concat}(\{\widetilde{\mathbf{Y}}_l^l, \widetilde{\mathbf{Y}}_{u,v_2}^l, \widetilde{\mathbf{Y}}_{u,v_1}^l\})$      ▷ **pseudo-labeling**
16:  $\hat{\mathbf{Y}}^l = \text{Softmax}(\mathcal{S}(\mathbf{Z}, C^l)/T)$     ▷ $\mathcal{S}(\cdot, \cdot)$ is cosine similarity function
17:  $\mathcal{L}_{tcd} = \sum_{l=1}^{L} \mathcal{L}_{ce}(\hat{\mathbf{Y}}^l, \widetilde{\mathbf{Y}}^l)$       ▷ **Eq.2:** $\mathcal{L}_{tcd}$
18: **end for**
  \*\*\*\*\*\*\*\*\*\*\*\*\**Taxonomic Context-based Prediction Alignment*\*\*\*\*\*\*\*\*\*\*\*\*\*\*
19: **for** layer $r = 1, ..., L$ and $r \neq$ target-grained layer $t$ **do**
20:  $\hat{\mathbf{Y}}^{t|r} = \text{Softmax}(\mathcal{S}(Z, C^r) \cdot \mathcal{S}(C^r, C^t)/T)$    ▷ **Eq.4**
21:  $\mathcal{L}_{tca} = \sum_{r=1, r \neq t}^{L} \mathcal{L}_{ce}(\hat{\mathbf{Y}}^{t|r}, \widetilde{\mathbf{Y}}^t)$     ▷ **Eq.5:** $\mathcal{L}_{tca}$
22: **end for**
23: $f_\theta^{(k+1)}, C^{(k+1)} \longleftarrow f_\theta^{(k)} - \gamma \triangledown_{f_\theta}(\mathcal{L}_{tcd} + \mathcal{L}_{tca}), C^{(k)} - \gamma \triangledown_C (\mathcal{L}_{tcd} + \mathcal{L}_{tca})$.
24: **end for**
25: **return** the trained encoder $f_\theta(\cdot)$ and target-grained $C^t$.

---

# H A Theoretical Interpretation for TIDA based Expectation-Maximization

In this section, we provide a more detailed theoretical interpretation for TIDA by extending hierarchical expectation-maximization (EM) algorithm [3].

We assume that the observed data $\mathcal{D}^l \cup \mathcal{D}^u$ are related to some latent variables sets which refer to the hierarchical prototypes $C = \{\{c_j^l\}_{i=1}^{n_l}\}_{l=1}^{L}$ above, where $n_l$ is the number of latent variable in set $C^l$, $l = 1, ..., L$ and $n_1 < n_2 < .. < n_l$. Here, we set $L$ as 3 and $C^2$ as the prototype set for target classification, i.e., $n_2 = |\mathcal{Y}^l| + |\mathcal{Y}^u|$.

For EM perspective, TIDA aims to maximizes the likelihood of the observed $N_l + N_u$ samples based on these hierarchical prototypes $C$, as follows:

$$\theta^*, C^* = \underset{\theta, C}{\arg\max} \sum_{i}^{N_l+N_u} \log p(x_i, \theta) = \underset{\theta, C}{\arg\max} \sum_{i}^{N_l+N_u} \sum_{C^l \in C} \log \sum_{c_j^l \in C^l} p(x_i, c_j^l; \theta). \quad \text{(I)}$$

According utilize Jensen's inequality [11], we can rewrite its surrogate function of Eq. I as follows:

$$\theta^*, C^* = \underset{\theta, C}{\arg\max} \sum_{i}^{N_l+N_u} \sum_{C^l \in C} \sum_{c_j^l \in C^l} Q(c_j^l) \log p(x_i, c_j^l; \theta) \quad \text{with} \quad Q(c_j^l) = p(c_j^l; x_i, \theta). \quad \text{(II)}$$

To sum up, TIDA aims to estimate the posterior class probability $Q(c_j^l) = p(c_j^l; x_i, \theta)$ on each hierarchy at the E-steps. Then, TIDA draw each sample to the prototype of its assigned cluster on

each hierarchy at the M-step by optimizing Eq. II with known $Q(c_j^l) = p(c_j^l; x_i, \theta)$. Moreover, to integrate structural priors into targets classification, TIDA improves the objective function on second hierarchy in the M-steps with aggregated prototypes. More details are shown as follows.

**E-step.** To estimate posterior class probability $Q(c_j^l) = p(c_j^l; x_i, \theta)$, following [16, 6], we use Sinkhorn-Koppn algorithm to assign samples to their clusters on each hierarchy, which prevents seen class from dominating the entire batch. Then we have the posterior class probability $Q(c_j^l) = p(c_j^l; x_i, \theta)$ as

$$Q(C_j^l) = p(c_j^l; x_i, \theta) = \begin{cases} 1 & if \quad x_i \in i\text{-th clutser} \\ 0 & else \end{cases} \quad (L = 1, 2, 3). \tag{III}$$

As we can see, the pseudo labels of samples $x_i$ equals $Q(c_j^l)$ on E-steps.

**M-step.** Given $Q(c_j^l)$ in Eq. III, we aims to maximize the surrogate function in Eq. II in M-step.

*(a) Separate Objective Functions for each Hierarchy.* First, we consider separate objective function for each hierarchy [11]. Here, we assume that the distribution around each prototype $c_j^l$ satisfy an isotropic Gaussian, so we have

$$p\left(x_i; c_j^l, \theta\right) = \exp\left(\frac{-\left(z_i - c_j^l\right)^2}{2(\sigma_j^l)^2}\right) / \sum_{j=1}^{n_l} \exp\left(\frac{-\left(z_i - c_j^l\right)^2}{2(\sigma_j^l)^2}\right) \tag{IV}$$

With normalized feature $z_i$ and prototype $c_j^l$, we assume the prior probability $p(c_j^l; \theta)$ for each $c_j^l$ as $\frac{1}{n_l}$, so we have

$$p(x_i, c_j^l; \theta) = p(x_i; c_j^l, \theta)p(c_j^l; \theta) = \frac{1}{n_l} \exp\left(\frac{2 - 2z_i \cdot c_j^l}{2(\sigma_j^l)^2}\right) / \sum_{j=1}^{n_l} \exp\left(\frac{2 - 2z_i \cdot c_j^l}{2(\sigma_j^l)^2}\right) \tag{V}$$

so we can write our objective function in M-steps as

$$\theta^*, C^* = \arg\max_{\theta, C^l} \sum_{i}^{N_l + N_u} \sum_{C^l \in C} \sum_{c_j^l \in C^l} -\log \frac{\exp\left(z_i \cdot c_s^{i,l}/\tau\right)}{\sum_{j=1}^{n_l} \exp\left(z_i \cdot c_j^l/\tau\right)}, \tag{VI}$$

where $c_s^{i,l}$ denotes $z_i$'s assigned prototype ($p(c_s^{i,l}; x_i, \theta) = 1$ and $c_s^{i,l} \in C^l$) and $\tau = 1 \propto (\sigma_j^l)^2$, which plays the role of temperature parameter. As we can see, Eq. VI is equivalent to the loss $\mathcal{L}_{tcd}$.

*(b) Constrain the Consistency across Hierarchies in Feature Space.* As [11] mentions, samples will be mapped to different mixtures of the von Mises-Fisher distributions on different hierarchy by optimizing Eq. VI. However, these distributions across different hierarchy may be inconsistent, which is harmful for performance. To tackle it and integrate structural priors into target tasks, we first obtain the aggregated latent variables $\widetilde{C}^{2|1}, \widetilde{C}^{2|3}$ with variables set $C^1, C^3$, then constrain the distribution around $\widetilde{C}^{2|1}, \widetilde{C}^{2|3}$ be consistent with the original one around $C^2$.

First, we establish the similarity relationship across adjacent hierarchy, *i.e.*, the mapping matrix $M^{r,2} = \mathcal{S}(C^{r\top}, C^2)$, where for $m^{r,2}[j,k] \in M^{r,2}$, $m^{2,r}[j,k] = \mathcal{S}(c_j^r, c_k^2)$. Then we obtained its aggregated variables set $\tilde{C}^{2|r}$ with latent variables $C^r$ as $\tilde{C}^{2|r} = \{\tilde{c}_k^{2|r}\}_{k=1}^{n_2}$, $\tilde{c}_k^{2|r} = \sum_{j=1}^{n_r} c_j^r m^{r,2}[j,k]$. Note, the variables set $\tilde{C}^{2|r}$ dynamically aggregates semantics from $r$-th hierarchy and builds communication among clusters across hierarchies, which facilitates aligning hierarchical semantics with target tasks.

Given aggregated variables, we then apply consistent constraints on the sample's distribution around $\tilde{C}^{2|r}$ and the original ones in the target hierarchy, where $r = 1, 3$. Specifically, we use aggregated

variables $\tilde{C}^{2|r}$ to calculate the objective function in Eq. VI, as follows:

$$
\begin{aligned}
\theta^*, C^* &= \arg\max_{\theta, C} \sum_i^{N_l+N_u} \sum_{r=1,3} -log \frac{\exp\left(z_i \cdot \tilde{c}_s^{i,2|r}/\tau\right)}{\sum_{k=1}^{n_2} \exp\left(z_i \cdot \tilde{c}_k^{2|r}/\tau\right)} \\
&= \arg\max_{\theta, C} \sum_i^{N_l+N_u} \sum_{r=1,3} -log \frac{\exp\left(z_i \cdot \sum_{j=1}^{n_r} c_j^r m^{r,2}[j,s]/\tau\right)}{\sum_{k=1}^{n_l} \exp\left(z_i \cdot \sum_{j=1}^{n_r} c_j^r m^{r,2}[j,k]/\tau\right)} \\
&= \arg\max_{\theta, C} \sum_i^{N_l+N_u} \sum_{r=1,3} -log \frac{\exp\left(\sum_{j=1}^{n_r} (z_i \cdot c_j^r)/\tau \cdot \mathcal{S}(c_j^r, c_s^2)\right)}{\sum_{k=1}^{n_l} \exp\left(\sum_{j=1}^{n_r} (z_i \cdot c_j^r)/\tau \cdot \mathcal{S}(c_j^r, c_k^2)\right)} \\
&= \arg\max_{\theta, C} \sum_i^{N_l+N_u} \sum_{r=1,3} -log \frac{\exp\left(\sum_{j=1}^{n_r} \mathcal{S}(z_i, c_j^r)/\tau \cdot \mathcal{S}(c_j^r, c_s^2)\right)}{\sum_{k=1}^{n_l} \exp\left(\sum_{j=1}^{n_r} \mathcal{S}(z_i, c_j^r)/\tau \cdot \mathcal{S}(c_j^r, c_k^2)\right)},
\end{aligned}
\tag{VII}
$$

where the index $s$ in $\tilde{c}_s^{i,2|r}$ corresponds to the assigned target-grained prototype $c_s^{i,2}$ for feature $z_i$ ($p(c_s^2; x_i, \theta) = 1$). As we can see, Eq. VII is equivalent to the loss $\mathcal{L}_{tca}$.

**Summary.** In this section, we provide another interpretation of TIDA, which can provide more insights into the nature of the learned structure priors. TIDA intrinsically: (a) builds hierarchical vMF distributions to cluster samples and discovery taxonomic context by optimizing Eq. VI; (b) applies the consistency constraint on the hierarchical vMF distributions to build communication and alignment across taxonomic context as Eq. VII. This involves the identification of patterns and relationships of data, and extracting discriminative features to enhance the performance.