# OpenReview forum: "Discover and Align Taxonomic Context Priors  for Open-world Semi-Supervised Learning"
_NeurIPS.cc/2023/Conference — NeurIPS 2023 poster_

### Official Review · Reviewer_uA1P · 2023-06-25

**Soundness:** 2 fair
**Presentation:** 3 good
**Contribution:** 3 good
**Rating:** 5
**Confidence:** 3

**Summary:**

This paper addresses the open-world semi-supervised learning (OSSL) problem and proposes taxonomic context priors discovering and aligning (TIDA), which considers the taxonomic hierarchy of classes.

Basically, the proposed method is based on [19], which assigns pseudo-labels to unlabeled samples by means of the Sinkhorn-Knopp algorithm. The proposed method is characterized by Taxonomic Context Discovery (TCD), which uses the sum of the losses at different hierarchical levels and Taxonomic Context-based Prediction Alignment (TCA), which adjusts the similarity between the feature and the class prototype by incorporating the weights based on the similarities between the prototypes at different hierarchical levels.

Experimental results on seven datasets show that the proposed method achieves better accuracy than existing methods.


**Strengths:**

* This paper proposes an OSSL method focusing on class hierarchy, which has not been considered in the past as far as I know. (However, as discussed below, similar ideas have already been considered in related problems.)

* The theoretical justification of the proposed method is discussed based on the principle of the EM algorithm (lower bound maximization of log-likelihood).

* Exhaustive experiments using seven datasets show that the proposed method is consistently more accurate than existing methods.


**Weaknesses:**

a. The idea of discovering unknown classes by focusing on the class hierarchy has been explored in the past [46], so the high-level novelty would be somewhat limited.

b. In this paper, the number of unknown classes is known, which is not realistic. The supplementary material shows the experimental results for the case where the number of unknown classes is unknown (Tables V and VI). However, only the results for the case with the best clustering accuracy are presented. I do not think this is realistic, since clustering accuracy is not always evaluable in practice.

c. The base networks used in the experiments are limited to ResNet-18/-50. Testing with more modern networks such as ViT and vision-language models would be nice to emphasize the effectiveness of the proposed method.

d. Table 3 shows that accuracy improves when both TCP and TCA are used, but always degrades when TCA is not used. The reason for this is not clear.

e. The results in Figure 4 suggest that the appropriate number of super-classes and sub-classes may differ depending on the dataset. Justification is needed.

f. Some typos:

L10: pIrors -> prIors

L199: Sinkhorn-Koppn -> Sinkhorn-Knopp


**Questions:**

Among the weaknesses I listed, (b) performance evaluation when the number of classes is unknown, (d) reasons for performance degradation when using TCP alone, and (e) discussion of the number of superclasses and subclasses for different datasets are of particular importance. I look forward to getting responses for these points from the authors.

**Limitations:**

Limitations are discussed in the paper.

---

> ### Author Rebuttal · Authors · 2023-08-05
>
> >**Q1:** Compare with DCCL[46].
> >
> > **A1:** **i).** Motivation: DCCL[46] aims to enhance representation learning by applying contrastive learning among dynamically estimated conceptions. In contrast, our TIDA aims to enhance pseudo label quality by hierarchically clustering samples and discovering multi-granularity concepts as auxiliary supervision.
> >
> > **ii).** Method: DCCL[46] alternately estimates underlying visual concepts and applies a dual-level contrastive learning. However, our TIDA proposes constructing a set of hierarchical prototypes in the latent space to discover the underlying taxonomic context priors. It also enforces consistency across hierarchical predictions and provides additional reliable supervision through taxonomic context-based prediction alignment. In fact, DCCL[46] only has a basic single-level supervision at a specific moment. Instead, our TIDA always maintains multi-granularity supervision as a taxonomic prior and ensures its consistency.
> >
> > **iii).** **TIDA outperforms DCCL[46] on all datasets**: In A3, we show that our TIDA clearly outperforms DCCL [46]. That is, jointly using multi-granularity supervision has not been studied in DCCL[46] and our TIDA is very different from DCCL[46].
>
> > **Q2:** When the class number is unknown, only the results for the case with the best clustering accuracy are presented. I do not think this is realistic, since clustering accuracy is not always evaluable in practice.
> >
> > **A2:** Sorry for the confusion. The reason for only presenting the case with the best clustering accuracy is to ensure a fair and comprehensive comparison. The strategy is widely used in previous works [1-3].
> >
> > Moreover, following your suggestion, we carry out comparisons with TRSSL[1] on CIFAR100 with more estimated number classes. The accuracy on all classes is reported, as shown below. Our TIDA always outperforms [1].
> > | Estimated number | 80 | 90 | 100 | 110 | 120
> > |:----: |:----: | :----:| :----:| :----:|:----:|
> > TRSSL[4] | 54.2 | 58.5 | 60.3 |56.9 | 54.1
> > Ours |61.1 | 53.4 | 65.3| 62.9 | 60.5
> >
> > [1] Towards realistic semi-supervised learning. In ECCV, 2022.
> >
> > [2] Generalized category discovery. In In CVPR, 2022
> >
> > [3] A unified objective for novel class discovery. In ICCV, 2021.
> >
> > **Q3:** Additional results using ViT16 as the backbone network.
> >
> > **A3:** Below, we compare our method with [4-8]. For a fair comparison, following [4-8], we also use ViT as the backbone and incorporate contrastive learning during training. We report the accuracy of all classes. Our method achieves the best results on all datasets. We have included these results in the revision.
> >
> > | Methods |CIFAR10 | CIFAR100 | Aircraft | SCar|
> > |:----: |:----: | :----:| :----:| :----:|
> > DCCL[4] | 96.3 | 75.3 | - |43.1 |
> > PromptCAL[5] | 97.9| 81.2 | 52.2 | 50.2
> > GCD[6] | 91.5 | 73.0 | 45.0 | 39.0
> > SimGCD[7] | 97.1 | 80.1 | 54.2 | 53.8
> > PIM[8] |94.7 |78.3 |- |43.1
> > Ours |**98.2** |**82.3** |**54.6** |**54.7**
> >
> >[4] Dynamic Conceptional Contrastive Learning for Generalized Category Discovery,CVPR 2023.
> >
> >[5] Promptcal: Contrastive affinity learning via auxiliary prompts for generalized novel category discovery, CVPR 2023.
> >
> >[6] Generalized category discovery, CVPR 2022.
> >
> >[7] A Simple Parametric Classification Baseline for Generalized Category Discovery, ICCV 2023.
> >
> >[8] Mutual information-based generalized category discovery, arXiv, 2022.
>
> >**Q4:** The reasons for performance degradation when using TCP alone.
> >
> >**A4:** The inconsistency among multi-granularity classification leads to hard-optimization problem when using TCP alone. Actually, TCP applies multiple classifiers to cluster samples under different granularities. It can be regarded as multi-task learning with a shared backbone. However, due to a lack of supervision (e.g., semantic names and attributes), when using TCP alone, the training objective among classifiers is inconsistent and the optimization on the sub- and sup-classifiers is completely unsupervised. This may have a negative impact on model optimization and consequently result in a performance degradation. To address this problem, we propose TCA to align the training objectives among classifiers and enable the sub- and sup-classifiers to benefit from the knowledge of labeled data on the target classifier.
>
> >**Q5:** The results in Figure 4 suggest that the appropriate number of super-classes and sub-classes may differ depending on the dataset. Justification is needed.
> >
> > **A5:** Following your suggestion, we have provided the justification of α and β on the other datasets, as shown below.
> > We can observe that the values of α and β that yield the best performance are similar for generic datasets (α = 0.2, β = 2.0) and fine-grained datasets (α = 0.4, β = 2.5) respectively. We thus use one dataset to determine the values of α and β for the generic (or fine-grained) setting and apply the same values of α and β to all generic (or fine-grained) datasets. Please refer to L234-L235 in the original paper.
> >
> > **i.** The ablation analysis for α.
> > | Dataset Type | Dataset | 0.1 | 0.2 | 0.4 | 0.6 | 0.8
> >| :----: | :----: | :----: | :----: | :----: | :----: | :----: |
> >Generic | CIFAR100 | 63.4| **65.3** | 64.7 | 61.3 | 59.3
> >Generic | Tiny ImageNet | 35.8| **37.2** | 35.2 | 34.2 | 33.9
> >Fine-Grained | Standard Cars | 64.2 | 65.2 | **66.1** | 62.7 | 62.1
> >Fine-Grained | Oxford-IIIT Pets | 58.4 | 59.4 | **59.9** | 58.6 | 54.6
> >
> > **ii.** The ablation analysis for β.
> >| Dataset Type | Dataset | 1.25 | 1.5 | 1.75| 2.0| 2.5| 3.0
> >| :----: | :----: | :----: | :----: | :----: | :----: | :----: | :----: |
> >Generic | CIFAR100 | 61.94 | 62.1 | 63.6 | **65.3** | 61.7| 61.8
> >Generic | Tiny ImageNet |33.9 | 34 | 35.6 | **37.2** | 36.7 | 34.2
> >Fine-Grained | Standard Cars |62.5 | 64.9 | 65.9| 65.4| **66.1** | 65.3
> >Fine-Grained | Oxford-IIIT Pets | 54.1|55.3 | 58.3 | **60.2** | 59.9 | 59.4

---

> > ### Comment · Reviewer_uA1P · 2023-08-16
> >
> > I would like to thank the authors for their answers to my questions. Most of my questions have been solved. I would expect that the interpretations presented in A4 should be included in the main body of the paper, as these are useful in understanding the motivation and the design policy of the method.

---

> > > ### Author Response · Authors · 2023-08-16
> > > **Thanks for your acknowledgment**
> > >
> > > Dear Reviewer uA1P,
> > >
> > > We greatly appreciate your helpful comments and your satisfaction with our responses! We will add the above important discussions (i.e. The reasons for performance degradation when using TCP alone) in the final manuscript and highlight them.
> > >
> > > Thanks again for your valuable suggestions and comments. We really enjoy communicating with you and appreciate your efforts.

---

### Official Review · Reviewer_Qvan · 2023-07-05

**Soundness:** 3 good
**Presentation:** 3 good
**Contribution:** 2 fair
**Rating:** 6
**Confidence:** 3

**Summary:**

Previous research has primarily focused on using pre-defined single-granularity labels as priors for recognizing novel classes. However, classes naturally adhere to a taxonomy, enabling classification at multiple levels of granularity and offering richer supervision through underlying relationships. To address this, this paper proposes TIDA (Taxonomic context pIrors Discovering and Aligning), a unified framework that leverages sample relationships across various levels of granularity. TIDA discovers multi-granularity semantic concepts as taxonomic context priors (e.g., sub-class, target-class, and super-class) to enhance representation learning and improve pseudo label quality. Extensive experiments on seven datasets demonstrate TIDA's significant performance improvement, achieving a new state-of-the-art.

**Strengths:**

1. The motivation is clear and convincing. It is reasonable that taxonomic context priors are helpful to extract discriminative features.
2. The writing is clear. I like the figures in this paper.
3. Experiments effectively verify the effectiveness of this work. Comprehensive experimental results are provided in the supplementary material.

**Weaknesses:**

1. The necessity of constructing taxonomic priors remains unclear. Various methods, such as using WordNet [1], can extract taxonomic priors among categories. I think that there is insufficient evidence to support the essentiality of constructing taxonomic priors in typical scenarios.

[1] I Am Going MAD: Maximum Discrepancy Competition for Comparing Classifiers Adaptively. ICLR 2020.

2. Some works, especially [2], on the taxonomic structure lack proper citation and comparison.

[2] Open-world Semi-supervised Novel Class Discovery. IJCAI 2023.

3. The Introduction section lacks high-level statements regarding the construction and correctness of taxonomic priors.  need to find these information in the Methodology section.

**Questions:**

1. Can pre-defined taxonomic priors be utilized? Some datasets already have defined taxonomic priors, while WordNet can be used [1] to extract such priors among categories.

[1] I Am Going MAD: Maximum Discrepancy Competition for Comparing Classifiers Adaptively. ICLR 2020.

2. Have other cluster algorithms been explored for obtaining taxonomic priors, besides the SK algorithm?

3. Could you provide further clarification on why this work is suitable for open-world learning? I can understand its benefits for generating high-quality pseudo labels in semi-supervised learning.

4. Can you compare with [2] in detail?

[2] Open-world Semi-supervised Novel Class Discovery. IJCAI 2023.

**Limitations:**

In conclusion, this paper exhibits a clear motivation and comprehensive experimental verifications. However, the necessity of constructing taxonomic priors remains a main concern. It would be beneficial if the authors could provide specific scenarios where the construction of taxonomic priors is indeed essential. In addition, the comparison with [2] is essential.

This paper fully considers potential negative societal impact of their work.

---

> ### Author Rebuttal · Authors · 2023-08-05
>
> > **Q1&Q4:** Can pre-defined taxonomic priors be utilized?
> >
> > **A1:** We **CANNOT** directly construct taxonomic priors using language models, such as the mentioned WordNet[1]. This because we have no knowledge about novel classes in open-world SSL (neither semantic names nor attributes).
> > Thus, to leverage taxonomic knowledge in open-world SSL, it is crucial to design a method that can automatically discover/construct the taxonomic prior without using any extra supervision. In this regard, we introduce TIDA, which is specifically designed for this purpose using a novel cross-hierarchical prediction alignment approach.
>
> >**Q2&Q7:** Compare with [2].
> >
> > **A2:** Thanks for bringing this work to my attention.
> >
> > **i).** Motivation: [2] focuses on designing multiple learnable prototypes and aims to progressively group similar prototypes to discover novel classes. In fact, the prototypes and prototype groups involved can only be considered as an instance hierarchy and target-classes hierarchy, but not as taxonomic priors (sub-classes, target-classes, super-classes). In contrast, our TIDA applies a consistently hierarchical cluster learning approach and aims to simultaneously discover multi-granularity concepts as auxiliary supervision to facilicate discoverying novel classes.
> >
> > **i):** Method: [2] proposes a bi-level contrastive learning method to preserve the pairwise similarity between the prototypes and the prototype group levels, aiming to enhance representation learning. Instead, we construct a set of hierarchical prototypes in the latent space to uncover the underlying taxonomic context priors. In addition, we enforce consistency across hierarchical predictions through the proposed taxonomic context-based prediction alignment.
> >
> > **iii).** As shown in the table below, **our TIDA outperforms [2] on all datasets**.
> >| |CIFAR10 | CIFAR100 | ImageNet-100
> >|:----: |:----: | :----:|:----:|
> > OpenNCD[2]|85.3 |41.2 |73.2
> > Ours |**93.8**|**65.3**|**77.6**
> >
> >We would like to reminder that [2] was released after the NeurIPS deadline (22 May 2023 vs 17 May 2023). We will compare it with [2] in the next version.
>
> > **Q3:** The Introduction section lacks high-level statements regarding the **construction** and **correctness** of taxonomic priors.
> >
> > **A3:** Thanks for this constructive suggestion. This is the high-level statements: " In this paper, we develop two modules: i) Taxonomic Context Discovery (TCD) module, which discovers the underlying taxonomic context priors by **constructing a set of hierarchical prototypes to cluster samples;** ii) Taxonomic Context-based prediction Alignment (TCA) module, which **enforces consistency across hierarchical predictions** to build reliable relationships between classes at various levels of granularity and provide additional supervision." We have included these statements in the revision.
>
> > **Q5:** Have other cluster algorithms been explored for obtaining taxonomic priors, besides the SK algorithm?
> >
> > **A5:** Good questions. In this paper, we adopt Sinkhorn as the online clustering algorithm due to its superiority, wide application, simplicity, and distribution-based learning strategy. Following your suggestion, we apply our approach on ORCA[25] and RankStats[24] and find that the model is hard to converge. The main reason may be that these two methods mainly reply on pair-wise similarity constraint to optimize classifier while our TIDA is constrained on the overall class distribution.
>
> > **Q6:** Could you provide further clarification on why this work is suitable for open-world learning?
> >
> > **A6:** The core challenge of open-set semi-supervised learning (OSSL) is the lack of supervision/labels (e.g., semantic names and attributes) for novel classes. To solve it, **the key is to exploit some priors (e.g. class distribution, pairwise similarity) as auxiliary supervision to recognize novel/seen classes,** which is the main goal of existing OSSL works.
> >
> > Here, we explain why this work is suitable for OSSL from three aspects:
> > **i).** The consistent taxonomic priors learned by TIDA provides more richer supervision:
> >
> > * Different from previous works with a single label hierarchy, TIDA first explores and utilizes consistent taxonomic priors as multi-granularity supervision. Given these priors, our TIDA is able to explore the relationship of samples under different granularity.
> > * For examples, TIDA enforce to distinguish these hard-to-distinguish samples ( e.g. *Telephone* and *Television*) in CIFAR100 with fine-grained priors, however TRSSL fails to classify them with single supervision. Moreover, TIDA use the coarse-grained priors to constrain samples from *Chair* and *Couch* to be closer and be far from *Baby*, as additional supervison. Refer to Figure IV,V in the supplementary.
> >
> > **ii).** TIDA provide a feasible way to automatically learn taxonomic priors without extra supervisions.
> > * We would like to emphasize that enforcing taxonomic consistency under open-world SSL is challenging due to the lack of information about novel classes, such as semantic names and attributes. Thus, we propose a unified OSSL framework that can discover taxonomic context priors and enforce cross-hierarchical consistency in an unsupervised manner.
> >
> > **iii).** More encouragingly, it can also serve as a visualization tool to promote the understanding of novel class, which have a far-reaching impact on the community.

---

> > ### Comment · Reviewer_Qvan · 2023-08-14
> >
> > Thank you for your comprehensive responses.
> >
> > I have two main concerns. Firstly, the necessity of constructing taxonomic priors needs clarification. A1's argument appears compelling. In the context of open-world learning, it seems to be inconvenient to utilize pre-defined class hierarchies.
> >
> > Secondly, the comparison with [2] is notable. The authors elucidate the distinctions between their work and [2] in A2.
> >
> > Additionally, I appreciate the authors' discussion on the core challenge of open-world learning, which offers valuable insights. I think that exploring the specific requirements of the mostly needed priors for open-world learning could be intriguing, because there are so many different priors to be used. For instance, identifying the essential properties that priors must fulfill for effective open-world learning (Personally, I am curious about this problem).
> >
> > In conclusion, the provided rebuttal has provided me with valuable information. Consequently, I have revised my rating to Weak Accept.

---

> > > ### Author Response · Authors · 2023-08-14
> > > **Thanks for your acknowledgment and improvement of rating**
> > >
> > > Dear Reviewer Qvan,
> > >
> > > We greatly appreciate your satisfaction with our responses ( particularly on the necessity of constructing taxonomic priors in our method, the difference to related work [2], and the discussion on the core challenge of open-world learning that we focused on solving), and very glad you increase the rating. We will include all the responses in the final manuscript. Thanks again for your great effort in handling our paper.

---

### Official Review · Reviewer_k9he · 2023-07-06

**Soundness:** 2 fair
**Presentation:** 2 fair
**Contribution:** 2 fair
**Rating:** 5
**Confidence:** 4

**Summary:**

This paper tackles open-world semi-supervised learning and proposes to use multi-granularity labels as taxonomic context priors to leverage hierarchical supervision to enhance representation learning and improve the quality of pseudo labels. A taxonomic context discovery module is used to construct hierarchical prototypes; a taxonomic context-based prediction alignment module is applied to enforce consistency across multi-granularity predictions.

**Strengths:**

1. The motivation and idea of this paper are clear and well-explained. Multi-granularity supervision is introduced to improve the representation of base and novel classes. The alignment among each granularity is enforced.

**Weaknesses:**

1. Additional baselines are needed for comparison:

[1] Pu N, Zhong Z, Sebe N. Dynamic Conceptional Contrastive Learning for Generalized Category Discovery[C]//Proceedings of the IEEE/CVF Conference on Computer Vision and Pattern Recognition. 2023: 7579-7588.

[2] Zhang S, Khan S, Shen Z, et al. Promptcal: Contrastive affinity learning via auxiliary prompts for generalized novel category discovery[C]//Proceedings of the IEEE/CVF Conference on Computer Vision and Pattern Recognition. 2023: 3479-3488.

[3] Vaze S, Han K, Vedaldi A, et al. Generalized category discovery[C]//Proceedings of the IEEE/CVF Conference on Computer Vision and Pattern Recognition. 2022: 7492-7501.

[4] Wen X, Zhao B, Qi X. A Simple Parametric Classification Baseline for Generalized Category Discovery[J]. arXiv preprint arXiv:2211.11727, 2022.

[5] Chiaroni F, Dolz J, Masud Z I, et al. Mutual information-based generalized category discovery[J]. arXiv preprint arXiv:2212.00334, 2022.

2. The actual number of classes must be known. Otherwise, it is difficult to determine how many classes should be defined for sub-class, target-class, and super-class, respectively.

3. The real composition of the sub-class and super-class may not go exactly as designed. It is difficult to guarantee each sub-class only contains samples of one target class, and each target-class only belongs to one super-class.

**Questions:**

1. The authors need to analyze the learned hierarchical tree of multi-granularity labels of each dataset. It could be the case that each super-class may contain different numbers of target-classes, and also each target-class may contain different numbers of sub-classes. Especially when the target-classes of one dataset are imbalanced in terms of similarity in semantic space.

2. The learning of multi-granularity pseudo-labels could be heavily-biased when the class-distribution is imbalanced and the semantic distance of each target-class is imbalanced.

**Limitations:**

The number of novel classes needs to be known. The class-distribution needs to be balanced. These assumptions make TIDA not robust for practical usage.

---

> ### Author Rebuttal · Authors · 2023-08-05
>
> We sincerely thank you for your valuable comments! Please find our detailed response below.
> >**Q1:** Compare with additional baselines.
> >
> >**A1:** Below, we compare our method with [1-5]. Our method achieves the best results on all datasets, as shown below.  For a fair comparison, following [1-5], we also use ViT as the backbone and incorporate contrastive learning during training. Here, we report the accuracy of all classes. We have included these results in the new version.
> >
> >| Methods|CIFAR10|CIFAR100|Aircraft| SCar|
> >|:----:|:----:|:----:|:----:|:----:|
> >DCCL[1]|96.3|75.3|-|43.1|
> >PromptCAL[2]|97.9|81.2|52.2|50.2
> >GCD[3]|91.5|73.0|45.0|39.0
> >SimGCD[4]|97.1|80.1|54.2|53.8
> >PIM[5]|94.7|78.3|-|43.1
> >Ours|**98.2**|**82.3**|**54.6**|**54.7**
>
> >**Q2:** The number of classes must be known.
> >
> > **A2:**  The number of classes does not need to be know.  On the one hand, the reason for assuming the number of novel classes as known in the main paper is to ensure a fair and comprehensive comparison with previous works [3,4,19,52] that all follow this assumption. On the other hand, we demonstrate that our TIDA still works well and achieves SOTA performance even when the number of classes is unknown, as shown in Tab. VI of the supplementary.
>
> >**Q3:** It is difficult to guarantee that each sub-class only contains samples of one target class, and that each target class only belongs to one super-class in real world.
> >
> >**A3:**  We agree with you that this assumption may not always hold true in the real world, but it is generally satisfied in most cases and beneficial to our TIDA. Thus, we propose TCA module and employ a relaxed strategy to enforcing prediction consistency across hierarchies and finally achieve high-quality consistency.
> >
> > Here, we discuss why the proposed assumption is beneficial to our TIDA:
> >
> > i). **The performance in TIDA is positively correlated with the consistency quality.** We calculate $\frac{1}{n_2}\sum_{i=1}^{n_2}\max_{j=1}^{n_1} A_{i,j}^{super}$ and $\frac{1}{n_2}\sum_{i=1}^{n_2} \max_{j=1}^{n_3} A_{j,i}^{sub}$ to quantitatively assess the consistencies of the super-target and sub-target on CIFAR100. Here, $A^{super}$ denotes the affinity matrix of the super-/target class level and the item $A^{sup}_{i,j}$ of $A^{sup}$ denotes the number of samples from the j-th target class that are classified into the i-th super-class. $A^{sub} \in R^{n_2 \times n_3}$ is similar for the sub-target levels (Refers to Fig.6 in the main paper). With the higher quality consistency, the TIDA achieves higher accuracy.
> > ||Super-Target Consistency|Sub-Target Consistency|Acc.
> >  |:----:|:----:|:----:|:----:|
> > TIDA |**72.3**|**69.2**|**65.3**
> > TIDA w/o TCA|65.7|61.3 |57.7
> >
> > ii). **The consistency assumption is necessary for our TIDA.** As shown in Tab.3 of our main paper, the taxonomic priors fails to improve performance without TCA. This is beacuse the inconsistency among multi-granularity classification leads to hard-optimization problem.
> >
> > In future, we will investigate the multiple mapping situation that better meet the real-world application.
>
> >**Q4:** The authors need to analyze the learned hierarchical tree for each dataset. Especially when the target classes of one dataset are imbalanced in terms of similarity in the semantic space.
>
> > **A4:** **i).** For each dataset in main paper, TIDA always captures the meaningful and balanced hierarchical structure. This is because TIDA assumes that both the superclass and subclass follow a class-balanced distribution. Please refer to the **Sec.E.2-E.4** in the supplementary.
> >
> > **ii).** For imbalanced similarity, TIDA still works well and tend to learn balanced distribution. But due to the imbalanced similarity, the learned super-classes repersents different grained semantics. For examples, a tiny dataset cotains 6 target classes (bed, chair, couch, wardrobe, baby, dog) and the number of super-classes is 3. The learned super-classes include #1 (chair, couch), #2 (wardrobe, bed), and #3 (baby, dog). The granularity of semantic expressed by super class #1/2 and #3 is different.
> >
> >  We also verify the effectiveness of TIDA in similarity-imbalanced setting. We randomly sample 90 classes from CIFAR100 to construct an imbalanced similarity in the semantic space (IS-CIFAR90), where the number of subclasses varies for each superclass. We report the average accuracy on all classes, as shown below. Results show that our TIDA still outperforms the TRSSL[52].
> >||TRSSL|Ours |
> >|:----:|:----:|:----:|
> >S-CIFAR90|51.8|**53.4**
>
> > **Q5:** The learning of pseudo-labels could be heavily-biased when the class-distribution is imbalanced and the semantic distance of target-class is imbalanced.
> >
> > **A5:** Good Question! The learning of multi-granularity pseudo-labels may be affected, but our TIDA still enables construct meaningful taxonomic priors to enhance performance with these imbalanced setting.
> >
> >  **i).** Imbalanced class distribution: Due to the assumption of class balance, the learning of multi-granularity pseudo-labels slight biased. **But the distribution of the pseudo-label on target class learned by TIDA is closer to the real distribution than the TRSSL [52].**
> >
> > Here, we conduct experiments following the TRSSL[52], which involves an imbalanced class distribution setting (more details refer to [52]). We report the average accuracy on all classes and the KL diversity of pseudo-label and Gound-Truth, as shown below. Results show that our TIDA still outperforms TRSSL.
> >
> > |Imbalance factors=10|Class imbalanced priors|KL|Acc.|
> > |:----:|:----: |:----:|:----:|
> > TRSSL|$\checkmark$ |0.0020|41.0
> > TRSSL|$\times$|0.0026|38.9
> > TIDA|$\times$|0.0017|**43.6**
> >
> > **ii).** Imbalanced semantic distance: The pseudo-labels learned by TIDA is balanced with this setting. Please refer to **A4** for details.
> >
> > **iii).** We will further investigate these two mentioned settings in future.

---

> > ### Comment · Reviewer_k9he · 2023-08-17
> >
> > I would like to thank the authors for their answers to my questions. Most of my questions have been resolved. For the listed results of A1, A3, A4, A5, how about the performance on "Seen" and "Novel"? It is important to analyze them separately.

---

> > > ### Author Response · Authors · 2023-08-18
> > > **Thanks for your acknowledgment and response to the further question**
> > >
> > > Dear Reviewer k9he,
> > >
> > > We greatly appreciate your satisfaction with our responses and your further suggestions. Below, we provide the comprehensive results along with corresponding analysis.
> > >
> > > >**Q1:** More comparison results with additional baseline.
> > > >
> > > >**A:** Below, we compare TIDA with [1-5] and report the accuracy of **Seen**, **Novel** and **All** classes. Here, we provide a thorough analysis for our TIDA as follows:
> > > >
> > > >  **i):** It is clear that TIDA achieves the best performance across all datasets and metrics, except for the CIFAR100-Seen accuracy and Aircraft-Novel accuracy which are slightly lower than PromptCAL. It is worth noting that PromptCAL is two stages while TIDA is end-to-end, which means that our TIDA outperforms PromptCAL in time-cost and computational cost.
> > > >
> > > >  **ii):**  It is surprising to observe that TIDA surpasses the state-of-the-art methods by **2.9%** and **1.2%** on CIFAR100-Novel and Scar-Novel, which demonstrates the superiority of the taxonomic context priors in our TIDA.
> > > >
> > > > These results demonstrate the superiority of our method over the state-of-the-art methods.
> > > >
> > > > A.1.1 Generic Dataset
> > > >
> > > > (Here, the numbers in bold and underline respectively represent the best performance and the inferior performance.)
> > > >
> > > >| Methods | CIFAR10-Seen | CIFAR10-Novel |CIFAR10-All | CIFAR100-Seen|CIFAR100-Novel|CIFAR100-All
> > > >|:----:|:----:|:----:|:----:|:----:| :----:|:----:|
> > > >DCCL[1]|96.5 |96.9 |96.3| 76.8 | 70.2 |75.3|
> > > >PromptCAL[2]| 96.6| **98.5**|$\underline{97.9}$| **84.2** | 75.3 |$\underline{81.2}$|
> > > >GCD[3]| **97.9** | 88.2 |91.5|76.2 |66.5 |73.0|
> > > >SimGCD[4]|95.1 |$\underline{98.1}$ |97.1|81.2 |$\underline{77.8}$|80.1|
> > > >PIM[5]| $\underline{97.4}$|93.3|94.7| **84.2** | 66.5 |78.3
> > > >Ours|**97.9**|**98.5** |**98.2**|$\underline{83.8}$|**80.7**|**82.3**
> > > >
> > > >A.1.2 Fine-Grained Dataset
> > > >
> > > >(Here, the numbers in bold and underline respectively represent the best performance and the inferior performance.)
> > > >
> > > >| Methods |Aircraft-Seen | Aircraft-Novel |Aircraft-All |SCar-Seen |SCar-Novel |SCar-All|
> > > >|:----:|:----:|:----:|:----:|:----:|:----:| :----:|
> > > >DCCL[1]|-|-|-|55.7|29.9|43.1|
> > > >PromptCAL[2]|52.2|**52.3**|52.2|70.1|40.6|50.2
> > > >GCD[3]|41.1|46.9|45.0|57.6|29.9|39.0
> > > >SimGCD[4]|$\underline{59.0}$|51.8|$\underline{54.2}$|$\underline{71.9}$|$\underline{45.0}$|$\underline{53.8}$
> > > >PIM[5]|-|-|-|66.9|31.6|43.1
> > > >Ours|**61.3**|$\underline{52.1}$|**54.6**|**72.3**|**46.2**|**54.7**
> > >
> > > > **Q3:** The performance in TIDA is positively correlated with the quality of consistency.
> > > >
> > > >**A:** To validate the effectiveness of our proposed consistency assumption, we investigate the relationship between layer consistency and performance by comparing **TIDA** and **TIDA w/o TCA**. As we can see, with higher quality consistency, TIDA achieves higher accuracies for both the **Seen** and **Novel**.
> > > >
> > > > ||Super-Target Consistency|Sub-Target Consistency|Seen|Novel|All
> > > >  |:----:|:----:|:----:|:----:|:----:|:----:|
> > > > TIDA w/o TCA |65.7|61.3 |69.9|54.4|57.7
> > > > TIDA |**72.3**|**69.2**|**73.3**|**56.6**|**65.3**
> > >
> > > > **Q4:** The comparison with baseline when target-classes are imbalanced semantic in distances.
> > > >
> > > >**A:** To verify the effectiveness of TIDA in a similarity-imbalanced setting, we compare our methods with TRSSL[52] on IS-CIFAR90. We report the average accuracy on Seen, Novel and ALL classes, as shown below.
> > > >
> > > >Results show that our TIDA surpasses TRSSL by **1.6%** and **5.1%** on the accuracy for All and Novel classes, respectively. Meanwhile, on the accuracy for Seen classes, TIDA is only lower than TRSSL by 0.5%. This indicates that, when target-classes are imbalanced in semantic distances, the learned consistent taxonomic context priors can effectively improve the recognition of novel classes classes without hampering the performance of seen classes.
> > > >
> > > >|S-CIFAR90|Seen|Novel|All|
> > > >|:----:|:----:|:----:|:----:|
> > > >TRSSL[52]|**62.7**|33.6| 51.8|
> > > >TIDA|62.2|**38.7**|**53.4**
> > >
> > > > **Q5:** The comparison with baseline when dataset is class distribution imbalanced.
> > > >
> > > >**A:** To verify the effectiveness of TIDA in imbalanced class distribution, we compare TIDA with TRSSL[52] on imbalanced CIFAR100.
> > > > Results in the table below show that our TIDA outperforms TRSSL across all metrics. The large improvement on novel classes verifies the benefit of our method in guiding the model to discover novel classes by taxonomic context priors.
> > > >
> > > > |Imbalance factors=10|Class imbalanced priors|KL|Seen|Novel|All
> > > > |:----:|:----: |:----:|:----:|:----:|:----:|
> > > > TRSSL[52]|$\checkmark$ |0.0020|52.9|27.4|41.0
> > > > TRSSL[52]|$\times$|0.0026|52.3|23.6|38.9
> > > > TIDA|$\times$|0.0017|**54.7**|**31.8**|**43.6**
> > >
> > > We hope that the comprehensive results and analysis could solve your remaining concerns. We will incorporate all of the discussions and experiments into the revision. Please feel free to let us know if you have any further questions. We would like to provide further explanation. Thank you once again for your time and valuable comments.

---

> > > > ### Comment · Reviewer_k9he · 2023-08-18
> > > >
> > > > Thanks for the authors' feedback. I would like to revise my rating to Borderline Accept.

---

> > > > > ### Author Response · Authors · 2023-08-18
> > > > > **Thanks for your acknowledgment and improvement of rating**
> > > > >
> > > > > Dear Reviewer k9he,
> > > > >
> > > > > We greatly appreciate your satisfaction with our responses, and very glad you increase the rating! We will add the above important discussions in the final manuscript and highlight them.
> > > > >
> > > > > Thanks again for your valuable suggestions and comments. We enjoy communicating with you and appreciate your efforts!

---

### Official Review · Reviewer_qAVC · 2023-07-07

**Soundness:** 3 good
**Presentation:** 2 fair
**Contribution:** 3 good
**Rating:** 5
**Confidence:** 3

**Summary:**

In this paper, a new Taxonomic context pIrors Discovering and Aligning (TIDA) which exploits the relationship of samples under various granularity is proposed. TIDA comprises two key components: i) A taxonomic context discovery module that constructs a set of hierarchical prototypes in the latent space to discover the underlying taxonomic context priors; ii) A taxonomic context-based prediction alignment module that enforces consistency across hierarchical predictions to build the reliable relationship between classes among various granularity and provide additions supervision.

**Strengths:**

1. The authors identify the importance of multi-granularity priors in the context of OSSL and introduce the taxonomic context priors for solving the OSSL problem.
2. A uniformed OSSL framework, which can discover taxonomic context priors without any extra supervision is proposed. With the proposed cross-hierarchical prediction alignment, the framework can effectively enhance the performance of the model.
3. This paper is easy to follow.

**Weaknesses:**

The hyper-parameter $\alpha$ and $\beta$ depends on the specific datasets and for different datasets, experiments need to be conducted to determine the value.

**Questions:**

Is it possible to set a init value of $\alpha$ and $\beta$, and then determine the specific value by dynamically decreasing or increasing?

**Limitations:**

The authors addressed the limitations.

---

> ### Author Rebuttal · Authors · 2023-08-05
>
> We sincerely thank you for your valuable comments! Please find our detailed response below.
> >**Q1:** The hyperparameters $\alpha$ and $\beta$ depend on the specific datasets, and experiments need to be conducted to determine their values for different datasets.
> >
> >**A1:** To avoid over-tuning parameters, we only conduct experiments on one generic dataset and one fine-grained dataset to determine the values of $\alpha$ and $\beta$. On the other hand, we find that the trends of $\alpha$ and $\beta$ are similar in all generic datasets or fine-grained datasets.
> >
> > Following your suggestion, we have provided the evaluation of $\alpha$ and $\beta$ on the other datasets, as shown below. We can observe that the values of $\alpha$ and $\beta$ that yield the best performance are similar for both generic datasets ( $\alpha$ = 0.2, $\beta$ = 2.0) and fine-grained datasets ($\alpha$ = 0.4, $\beta$ = 2.5) respectively. We thus use one dataset to determine the values of $\alpha$ and $\beta$ for the generic (or fine-grained) setting and apply the same values of $\alpha$ and $\beta$ to all generic (or fine-grained) datasets. Please refer to L234-L235 in the original paper.
> >
> >**i. The ablation analysis for $\alpha$**
> >| Dataset Type |  Dataset | 0.1 | 0.2 | 0.4 | 0.6 | 0.8
> >| :----: | :----: | :----: | :----: | :----: | :----: | :----: |
> >Generic | CIFAR100 | 63.4| **65.3** | 64.7 | 61.3 | 59.3
> >Generic |  Tiny ImageNet | 35.8| **37.2** | 35.2 | 34.2 | 33.9
> >Fine-Grained  |  Standard Cars | 64.2 | 65.2 | **66.1** | 62.7 | 62.1
> >Fine-Grained  |  Oxford-IIIT Pets | 58.4 | 59.4 | **59.9** | 58.6 | 54.6
> >
> >**ii. The ablation analysis for $\beta$**
> >| Dataset Type |  Dataset | 1.25 | 1.5 | 1.75| 2.0| 2.5| 3.0
> >| :----: | :----: | :----: | :----: | :----: | :----: | :----: | :----: |
> >Generic | CIFAR100 | 61.9 | 62.1 | 63.6 | **65.3** | 61.7| 61.8
> >Generic |  Tiny ImageNet |33.9 | 34.0 | 35.6 | **37.2** | 36.7 | 34.2
> >Fine-Grained  |  Standard Cars |62.5 | 64.9 | 65.9 | 65.4| **66.1** | 65.3
> >Fine-Grained  |  Oxford-IIIT Pets | 54.1|55.3 | 58.3 | **60.2** | 59.9 | 59.4
>
> >**Q2:** Is it possible to dynamically decrease or increase the value of $\alpha$ and $\beta$ during training?
> >
> > **A2:** Good question. Indeed, dynamically changing the values of $\alpha$ and $\beta$ during training can pose a **challenging optimization problem** for the model.
> > * Specifically, our TIDA applies two additional classifiers to classify samples at the super and sub-class levels, respectively. The shapes/dimensions of these classifiers are determined by the values of $\alpha$ and $\beta$.
> > * If we dynamically alter $\alpha$ and $\beta$ during training, we need to reinitialize the parameters after each change. This prevents the model from effectively utilizing the knowledge acquired by the super- and sub-class level classifiers during previous training iterations.
> > * Furthermore, the constraint between the continuously learned target-level classifier and the repeatedly re-initialized super- and sub-class level classifiers will also hinder the convergence of the model.
> >
> > Below, we conduct an experiment to dynamically change the values of $\alpha$ and $\beta$ on CIFAR100. We set the initial values of $\alpha$ and $\beta$ to 1. Then, we decrease $\alpha$ by 0.2 (or increase $\beta$ by 0.25) every 40 epochs, for a total of 200 epochs. We can find that the results, particularly on the novel class, are worse. Nevertheless, the above optimization problem only appears when we use the static network, and we will explore this interesting idea in the future.
> >
> >|  | seen | novel | all
> >| :----: | :----: | :----: | :----: |
> >| TRSSL  | 68.5 | 52.1 | 60.3
> >|Our w/ fixed $\alpha$/$\beta$ | **73.3** | **56.6** | **65.7** |
> >|Our w/ Dynamical $\alpha$/$\beta$ | 40.1 | 8.9 | 5.4 |

---

> > ### Comment · Reviewer_qAVC · 2023-08-18
> >
> > I would like to appreciate the authors taking time to answer questions during the rebuttal. My questions and concerns are addressed. I keep my score as borderline accept.

---

> > > ### Author Response · Authors · 2023-08-19
> > > **Thanks for your acknowledgment**
> > >
> > > Dear Reviewer qAVC,
> > >
> > > We greatly appreciate your helpful comments and your satisfaction with our responses! We will add the above important discussions in the final manuscript and highlight them. We really enjoy communicating with you and appreciate your efforts.

---

### Author Rebuttal · Authors · 2023-08-07

We sincerely thank the ACs and reviewers for their great effort in handling our paper.

We have appropriately addressed all concerns raised by the reviewers. These include providing more comparisons with recent baselines/backbones (Reviewer #k9he, #Qvan, #uA1Ps), conducting more ablation studies on the hyper-parameters $\alpha$ and $\beta$ (Reviewer #qAVC, #Qvan), discussing the realistic setting of our method (Reviewer #k9he,#uA1Ps), more quantitative evaluations of the learned hierarchical tree (Reviewer #k9he), and providing a clearer explanation of our motivation (Reviewer #Qvan).

Paper strengths acknowledged by reviewers:
+ The motivation and idea are **clear and convincing**. It is **reasonable** to assume that taxonomic context priors are helpful in extracting discriminative features. (Reviewer #qAVC, #k9he, #Qvan, #uA1Ps)

+ Experiments **effectively validate** the effectiveness of this work. **Comprehensive experimental results** are provided in the supplementary material. (Reviewer #qAVC, #k9he, #Qvan, #uA1Ps)

+ The paper is **well-written** and easy to understand. (Reviewer #qAVC, #k9he, #Qvan)

+ This paper proposes an OSSL method that focuses on class hierarchy, which has **not been considered** in the past. (Reviewer #uA1Ps)

+ The **theoretical justification** of the proposed method is discussed based on the principles of the EM algorithm (Reviewer #uA1Ps)

We expect ACs and reviewers to fully consider the following factors when making the final decision: (1) a novel taxonomic consistency learning framework for open-world SSL, with significant effectiveness and theoretical justification confirmed by the reviewers, (2) comprehensive responses to all the reviewers' comments, and (3) the source code opened to the reviewers (in the supplementary material).

Please let us know if you have any additional questions or concerns. We are happy to provide clarification.

Authors of Submission #2448

---

### Comment · Area_Chair_pWjD · 2023-08-20
**Kindly remind all reviewers to evaluate authors' responses**

Dear Reviewers,

Hope this message finds you well. We appreciate your dedicated reviews for NeurIPS 2023.
Kindly note that authors have responded to your feedback. Your prompt evaluation of their responses is crucial for finalizing papers.
Please log in and assess their responses at your earliest convenience. Your timely input ensures paper improvements and conference quality.
Access the system to review responses. Reach out if you need assistance.


Best regards,

Area Chair

---

### Decision · Program_Chairs · 2023-09-21

**Decision:**

Accept (poster)

**Comment:**

In this paper, the authors introduced TIDA, a framework for Open-world Semi-Supervised Learning (OSSL), which leverages multi-granularity semantic concepts to enhance representation learning and improve pseudo label quality. The extensive experiments demonstrate TIDA's significant accuracy improvements, surpassing the current state of the art, particularly by 6.9% on TinyImageNet. The experimental results and motivation in the paper are explained in detail, and the proposed method demonstrates a certain level of innovation. Following the rebuttal phase, all the reviewers have provided positive evaluations. Therefore, the AC recommends accepting this paper.